# STAR-VAE: Structured Topology-Aware Regularization for Audio Reconstruction and Generation

Huadai Liu [1 2]   Wen Wang [2]   Kaicheng Luo [2]   Qian Chen [2]   Xiangang Li [2]   Wei Xue [1]

## Abstract

Continuous Variational Autoencoders (VAEs) serve as the fundamental continuous tokenizer for modern neural audio generation systems, enabling high-fidelity reconstruction while providing a compact, smooth latent space for downstream generative priors. However, continuous VAEs face a fundamental conflict when balancing *compression rate*, *reconstruction fidelity*, and *latent space topology*—a challenge we formalize as the **Rate-Distortion-Regularity Trilemma**. This trilemma stems from a critical *topological mismatch*: the prevailing isotropic Gaussian prior in standard VAEs imposes a *flat* latent geometry that fails to accommodate audio's *hierarchical* nature, where low-frequency components are structured and compressible while high-frequency components are stochastic and incompressible, leading to *disordered information packing* where crucial semantic features are randomly interleaved with high-frequency noise. To resolve this challenge, we propose **Structured Topology-Aware Regularization (STAR)**, a general training strategy that reshapes latent space geometry by imposing a growth-based constraint field, routing structural and textural information into channel subspaces with matching capacities. STAR is applicable to any VAE architecture and effectively resolves the trilemma, as demonstrated in CNN-based VAEs. To fully exploit STAR's potential, we present **STAR-VAE**, combining STAR with a hybrid CNN-Mamba architecture that synergizes local feature extraction with linear-complexity global context modeling, achieving state-of-the-art performance. We further propose **STAR-Gen**, an LLM-based Flow Matching framework that leverages STAR-VAE's structured latent space for high-fidelity generation without suffering from vector quantization artifacts. Empirical results demonstrate that STAR-VAE successfully resolves the trilemma, achieving state-of-the-art reconstruction fidelity and enhanced semantic information preservation across diverse audio domains. The structured latent space improves both traditional diffusion models and our **STAR-Gen** paradigm, achieving state-of-the-art performance in text-to-audio generation. The project page is available at https://STAR-VAE.github.io.

## 1. Introduction

The landscape of audio generation has been revolutionized by the advent of Latent Diffusion Models (LDMs) (Rombach et al., 2022; Liu et al., 2023a; 2024a) and Flow Matching transformers (Liu et al., 2025a; Cheng et al., 2025; Liu et al., 2025d;b). However, the generative fidelity of these systems is fundamentally upper-bounded by the quality of the underlying latent representation (Zheng et al., 2025; Shi et al., 2025). Serving as the prevalent continuous tokenizer, a Variational Autoencoder (VAE) (Vahdat & Kautz, 2020; Pinheiro Cinelli et al., 2021) typically bears a dual responsibility: it must act as a high-fidelity *signal reconstructor* to preserve acoustic details, while simultaneously functioning as a *manifold regularizer* to provide a compact, smooth latent space for downstream generative priors.

Despite their success in image synthesis (Podell et al., 2023; Esser et al., 2024), standard VAE methods face severe challenges when adapted to high-dimensional, temporal audio signals. We identify a fundamental conflict—the **Rate-Distortion-Regularity Trilemma**—stemming from a critical *topological mismatch*: audio signals possess a strong spectral hierarchy (deterministic global structures to stochastic local textures, as in Figure 1 left), yet the prevailing isotropic Gaussian prior (Liu et al., 2023a; Evans et al., 2025; Wang et al., 2025) imposes a "flat" latent geometry, *treating all channels as equally informative*. This mismatch leads to **disordered information packing**, where crucial semantic features are randomly interleaved with high-entropy noise, resulting in suboptimal compression and chaotic latent topologies (visualized in Figure 1 top right).

[1]Hong Kong University of Science and Technology [2]Tongyi Fun Team, Alibaba Group. Correspondence to: Wei Xue <weixue@ust.hk>.

*Proceedings of the 43rd International Conference on Machine Learning*, Seoul, South Korea. PMLR 306, 2026. Copyright 2026 by the author(s).

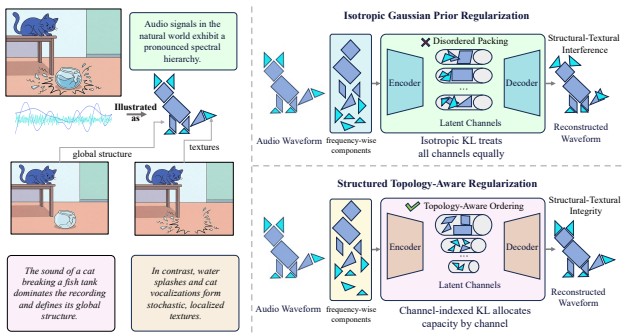

*Figure 1.* Conceptual illustration. **Left** visualizes audio's intrinsic hierarchy using a metaphor where geometric blocks represent **Global Structure** (e.g., impact sounds) and scattered fragments represent **Stochastic Textures** (e.g., splashes). **Top Right:** Standard Isotropic VAEs assign *equal* capacity to all channels, leading to disordered feature interleaving and information loss. **Bottom Right:** The proposed **Structured Topology-Aware Regularization** imposes a *Capacity Gradient (tapered channels)* that implicitly sorts features, routing complex structure to high-capacity channels and stochastic details to low-capacity ones.

The dominant architectures for Audio VAEs are built upon Convolutional Neural Networks (CNNs) (O'shea & Nash, 2015). While CNNs excel at extracting local spectral features, their limited receptive fields restrict their ability to model long-range temporal dependencies and global semantic structures. A natural evolution is to integrate advanced sequence models, such as State Space Models (SSMs) (Gu et al., 2020; 2021) (e.g., Mamba (Gu & Dao, 2024)), which offer linear-complexity global context. However, naively integrating such high-capacity encoders into the existing isotropic framework exacerbates the trilemma as we observe a counter-intuitive phenomenon termed **"Reconstruction Drift"**: powerful encoders such as Mamba, driven to minimize the uniform Kullback-Leibler (KL) penalty, spontaneously prioritize low-entropy structural information at the expense of high-entropy textural fidelity, producing "hollow" reconstructions—semantically coherent but texturally void.

To resolve the Rate-Distortion-Regularity Trilemma, we propose **Structured Topology-Aware Regularization (STAR)**, a general training strategy that realigns the bottleneck capacity with the intrinsic hierarchy of audio. Instead of a uniform penalty, STAR imposes a gradient of constraints across the latent channels. This inductive bias enforces a spectrally ordered latent space: complex, low-frequency structural information is naturally routed to "high-capacity" channels (low KL penalty), while high-entropy stochastic details are preferentially allocated to "low-capacity" channels (high KL penalty), effectively resolving the trilemma by organizing information according to its semantic importance.

Crucially, **STAR is a general regularization technique applicable to any VAE architecture, not limited to sequence models**. Our empirical analysis verifies that STAR

substantially improves reconstruction fidelity when applied to standard CNN-based VAEs, confirming its effectiveness as a general solution to the trilemma. To further demonstrate STAR's capability to unlock advanced sequence modeling, we propose **STAR-VAE**, which combines STAR with a hybrid CNN-Mamba architecture that synergizes local feature extraction with linear-complexity global context modeling. This combination achieves state-of-the-art (SOTA) performance, showcasing how STAR enables the safe deployment of high-capacity sequence models that would otherwise suffer from reconstruction drift under isotropic constraints. Our main contributions are as follows:

- We formalize the **Rate-Distortion-Regularity Trilemma** in audio VAEs and identify the isotropic Gaussian prior as the root cause of "disordered information packing".
- We propose **Structured Topology-Aware Regularization (STAR)**, a general regularization technique that induces a hierarchical channel ordering, effectively aligning latent capacity with signal information density and resolving the Rate-Distortion-Regularity Trilemma.
- Based on STAR, we introduce **STAR-VAE**, an enhanced VAE architecture that combines STAR regularization with a hybrid CNN-Mamba backbone, leveraging linear-complexity State Space Models to capture long-term audio dynamics. We propose **STAR-Gen**, an LLM-based Flow Matching framework that leverages the *structured* latent space of STAR-VAE to enable high-fidelity generation without suffering from vector quantization artifacts.
- Extensive experiments demonstrate the universal effectiveness of STAR across diverse VAE architectures (CNN-based and hybrid) and audio domains (sound and music). STAR-VAE achieves SOTA reconstruction fidelity, substantially outperforming strong baselines in semantic preservation and latent regularity. STAR-VAE's structured latent space benefits both traditional diffusion models and our STAR-Gen paradigm, achieving SOTA performance in text-to-audio generation. Ablation studies confirm that STAR induces hierarchical information organization, prevents Reconstruction Drift in high-capacity encoders, and the hybrid CNN-Mamba architecture optimally balances global context modeling with linear efficiency.

## 2. Preliminaries and Problem Formulation

In this section, we formalize the framework of audio VAEs and analyze the theoretical limitations imposed by their standard isotropic Gaussian prior.

### 2.1. Audio VAEs

Let $x \in \mathbb{R}^{T \times A}$ denote the raw audio waveform, where $T$ is the number of time samples and $A$ is the number of audio channels. A variational audio compression model consists of an encoder $\mathcal{E}_\phi$ parameterized by $\phi$ that maps $x$ to a pos-

terior distribution $q_\phi(z|x)$ in the latent space $z \in \mathbb{R}^{T' \times C}$, where $T'$ is the compressed temporal length and $C$ is the number of latent channels. A decoder $\mathcal{G}_\theta$ parameterized by $\theta$ reconstructs the waveform $\hat{x} \sim p_\theta(x|z)$. Distinct from standard VAEs used in simple image tasks, modern audio VAEs optimize a compound objective (Evans et al., 2025; Zeghidour et al., 2021; Défossez et al., 2022):

$$\mathcal{L}_{\text{Total}} = \mathcal{L}_{\text{Rec}}(x, \hat{x}) + \lambda_{\text{Adv}}\mathcal{L}_{\text{Adv}}(x, \hat{x}) + \beta\mathcal{L}_{\text{KL}}(q_\phi||p) \quad (1)$$

where $\mathcal{L}_{\text{Rec}}$ is the reconstruction loss, $\mathcal{L}_{\text{Adv}}$ combines adversarial and feature matching losses, and $\mathcal{L}_{\text{KL}}$ is the regularization weighted by $\beta$.

**Reconstruction Loss ($\mathcal{L}_{\text{Rec}}$):** To capture both temporal and spectral fidelity, we employ the Multi-Resolution STFT loss (Yamamoto et al., 2020), which explicitly models frequency-domain characteristics and better preserves perceptual audio quality compared to time-domain losses. This loss minimizes the $L_1$ distance and spectral convergence across $M$ FFT windows of different sizes:

$$\mathcal{L}_{\text{Rec}} = \sum_{i=1}^{M} \left(\|S_i(x) - S_i(\hat{x})\|_1 + \|\log S_i(x) - \log S_i(\hat{x})\|_2\right)$$
$$(2)$$

where $M$ is the number of resolution scales and $S_i(\cdot)$ denotes the Short-Time Fourier Transform (STFT) at the $i$-th resolution scale.

**Adversarial Loss ($\mathcal{L}_{\text{Adv}}$):** To alleviate the "oversmoothing" artifact typical of MSE-based reconstruction, a discriminator is introduced with adversarial and feature matching objectives to improve perceptual quality.

**Regularization ($\mathcal{L}_{\text{KL}}$):** A bottleneck constraint is imposed to limit the information capacity of $z$, forcing the encoder to learn a compact representation. The standard formulation minimizes the KL divergence between the posterior $q_\phi(z|x)$ and a fixed prior $p(z)$.

### 2.2. The Isotropic Prior Hypothesis

The critical flaw in current paradigms lies in the formulation of the regularization term. The standard prior is assumed to be an *isotropic* unit Gaussian, $p(z) = \mathcal{N}(0, I)$. Therefore, the KL penalty is applied *uniformly* across all $C$ channels:

$$\mathcal{L}_{\text{KL}} = \frac{1}{2}\sum_{c=1}^{C}\sum_{t=1}^{T'}\left(\mu_{c,t}^2 + \sigma_{c,t}^2 - \log\sigma_{c,t}^2 - 1\right) \quad (3)$$

where $\mu_{c,t}$ and $\sigma_{c,t}$ denote the mean and standard deviation of the posterior distribution $q_\phi(z_{c,t}|x)$ for channel $c$ at time step $t$. This formulation implies a fundamental hypothesis: **All latent channels possess equal capacity and semantic relevance**. The optimization landscape is "flat" with respect to the channel index $c$, penalizing information storage identically across all channels.

### 2.3. The Rate-Distortion-Regularity Trilemma

Audio signals exhibit a **Spectral Hierarchy**: information is non-uniformly distributed across frequencies. Low-frequency components (e.g., fundamental frequency, rhythm) are highly structured and compressible (Low Entropy), while high-frequency components (e.g., transients, noise) are stochastic and incompressible (High Entropy). We argue that the Isotropic Prior Hypothesis creates a fundamental topological mismatch with the audio signal, leading to a conflict we term the **Rate-Distortion-Regularity (R-D-R) Trilemma**, as follows.

- **Distortion vs. Rate:** High-fidelity reconstruction requires capturing high-entropy spectral details. Under a uniform KL penalty, the cost of encoding these stochastic bits equally across all channels is prohibitively high.
- **Distortion vs. Regularity:** To bypass the uniform penalty and preserve details, the encoder will be forced to deviate significantly from the isotropic prior, creating a "jagged" posterior distribution with high variance. This destroys the Regularity of the latent space—the smoothness and predictability required for downstream generation.

Therefore, the core limitation of standard VAEs is **lack of Latent Ordering**: the isotropic prior fails to provide a "shelving system" that aligns with the intrinsic hierarchy of audio data.

**The Consequence of the Trilemma: Disordered Information Packing.** In an isotropic bottleneck, the model lacks an inductive bias to organize information and leads to **Disordered Information Packing**: it packs low-entropy structure and high-entropy noise indiscriminately into any available channel. For a downstream generative model, this behavior results in a *noisy optimization landscape*: the model cannot rely on specific latent dimensions to carry consistent semantic levels. It must learn to denoise structure and texture simultaneously across all dimensions, significantly increasing the complexity of the generation task. We provide empirical evidence of this disordered information packing through channel-wise KL divergence analysis shown in Figure 3(a) in Section 4, contrasting the structured hierarchy induced by STAR with the chaotic, multi-modal distribution of information in isotropic baselines.

## 3. Methodology

In this section, we introduce our proposed framework to resolve the Rate-Distortion-Regularity Trilemma. We first present **Structured Topology-Aware Regularization (STAR)**, a general training strategy that induces a hierarchical topology in the latent space. We then describe the hybrid CNN-Mamba architecture that, when synergistically combined with STAR, forms the **STAR-VAE** model (illustrated on the left of Figure 2). Finally, we present **STAR-Gen**, a novel LLM-based Flow Matching paradigm that leverages

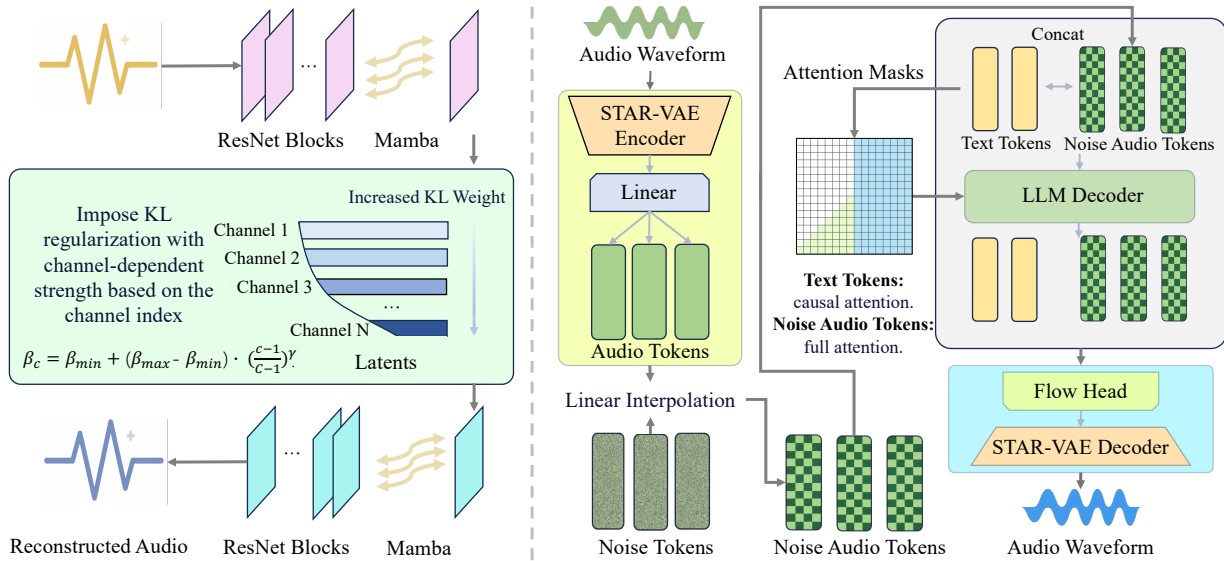

*Figure 2.* Overview of the STAR-VAE (Left) and STAR-Gen (Right) framework. **Left:** The STAR-VAE encoder projects raw audio into a hierarchically organized latent space, explicitly sorting features by information density via the STAR regularization. **Right:** The generative model, STAR-Gen, represents a paradigm shift by adapting an LLM Decoder backbone for continuous flow matching. Unlike discrete autoregressive models, it utilizes the frozen STAR latents as continuous targets and predicts the vector field from noise, employing a hybrid attention mechanism to integrate text conditioning.

STAR-VAE's structured latent space for high-fidelity audio generation (illustrated on the right of Figure 2).

### 3.1. Structured Topology-Aware Regularization (STAR)

To address the disordered information packing caused by isotropic priors, STAR abandons the uniform KL penalty in favor of a **Structured Constraint Field**.

**Design Philosophy: The Capacity Gradient.** Our primary objective is to create a "Capacity Gradient" that aligns the latent space with the audio signal's intrinsic hierarchy. We divide the latent space into two continuous functional zones. The **High-Capacity Zone (Low $\beta$ in Eq. 1)** is a "Safe Harbor" for non-Gaussian, deterministic information (Structure), where low penalty allows storing complex semantic dependencies. The **Low-Capacity Zone (High $\beta$)** is a "Noise Floor" for stochastic, high-entropy information (Texture), where high penalty forces adherence to the standard Gaussian prior, suitable for white-noise-like residuals or channel pruning.

**Deriving the Optimal Constraint Curve.** The core challenge lies in defining the transition between these zones. A **Step Function** imposes an artificial binary distinction, ignoring the continuous spectrum of audio features. A **Linear Function** offers a gradient but lacks the flexibility to model non-linear information decay. From an information-theoretic perspective, natural signals (including audio spectrograms) exhibit power-law decay in their energy spectrum (Zipf's Law or $1/f$ noise) (Field, 1987; Voss & Clarke, 1975), implying that the "value" of the $k$-th latent factor drops polynomially. To match this distribution, the capacity

allocation should follow a power law.

**The Gamma-Growth Implementation.** Based on this theoretical insight, we parameterize the channel-wise penalty vector $\boldsymbol{\beta} \in \mathbb{R}^C$ using a **Gamma-Growth Function**:

$$\beta_c = \beta_{\min} + (\beta_{\max} - \beta_{\min}) \cdot \left( \frac{c-1}{C-1} \right)^{\gamma} \quad (4)$$

where $\gamma > 0$ (the spectral curvature) controls the *information distribution density*.

From an information-theoretic standpoint, audio signals exhibit a **Heavy-Tailed Spectral Distribution** (Voss & Clarke, 1975): the vast majority of perceptual information (e.g., fundamental frequencies, formants) is concentrated in the low-to-mid frequency bands, while high-frequency bands contain sparse, stochastic details. To match this intrinsic "Information Mass", the latent space requires a larger proportion for "Low-Penalty" channels to accommodate the dense structural content.

Therefore, we choose a **Convex Allocation** ($\gamma > 1$). By slowing the growth of $\beta_c$ in the lower channel indices, we effectively widen the "Safe Harbor" (Low-Penalty Zone), allocating more channel capacity to the information-rich structural components. Conversely, a Concave ($\gamma < 1$) or Linear ($\gamma = 1$) allocation would prematurely penalize these critical features, forcing the model to discard structural information or displace it into high-penalty zones, thereby re-introducing disorder. Convex allocation allows STAR to optimally "water-fill" the latent space according to the signal's natural information density. We provide empirical

validation of this design choice through ablation studies comparing different growth functions and $\gamma$ values in Appendix C.1.

**Inductive Ordering.** With STAR regularization, only the third term $\beta\mathcal{L}_{\text{KL}}$ in Eq. 1 is replaced by

$$\mathcal{L}_{\text{STAR}} = \sum_{c=1}^{C} \beta_c \cdot D_{\text{KL}}(q_\phi(z_c|x)||\mathcal{N}(0,1)) \qquad (5)$$

while the reconstruction loss $\mathcal{L}_{\text{Rec}}$ and adversarial loss $\mathcal{L}_{\text{Adv}}$ remain unchanged. By solving this anisotropic optimization problem, the encoder implicitly learns to sort features by their information density, routing global structure to low channel indices (forming the **Structure Subspace**) and local texture to high channel indices.

### 3.2. Hybrid CNN-Mamba Architecture

With STAR providing a robust topological constraint, we can effectively deploy high-capacity sequence models to capture global audio dynamics without suffering from Reconstruction Drift. The hybrid architecture component of **STAR-VAE** combines the local feature extraction of CNNs with the long-range context modeling of Selective State Space Models (Mamba) (Gu & Dao, 2024).

**Hybrid Encoder Design.** The encoder $\mathcal{E}_\phi$ is structured to progressively abstract the signal from local waveforms to global semantics.

(1) **Local Downsampling (CNN):** The raw waveform is processed by a stack of strided ResNet blocks (He et al., 2016). This stage efficiently extracts high-frequency spectral details and compresses the temporal resolution, overcoming the quadratic cost of processing raw samples directly.

(2) **Global Context (Mamba):** To address the limited receptive field of CNNs, the compressed feature sequence is passed through a bidirectional Mamba backbone. Unlike standard Transformers with quadratic complexity, Mamba utilizes a **Selective State Space Mechanism** to model global dependencies with linear complexity $\mathcal{O}(T)$. Formally, each Mamba block operates as a discretized continuous system mapping input $x(t)$ to output $y(t)$ via a hidden state $h(t)$:

$$h'(t) = \mathbf{A}h(t) + \mathbf{B}x(t), \quad y(t) = \mathbf{C}h(t) \qquad (6)$$

where $\mathbf{A}$ denotes the state transition matrix, $\mathbf{B}$ and $\mathbf{C}$ denote the input and output matrix, respectively. Through a discretization process (Zero-Order Hold) with a timescale parameter $\Delta$, these parameters become input-dependent functions $\mathbf{B}(x), \mathbf{C}(x), \Delta(x)$. This mechanism allows the model to selectively propagate relevant structural information while filtering out irrelevant noise. **This content-aware selectivity is crucial for maximizing the utility of the**

**Structure Subspace (low-index channels storing global structure) induced by STAR**.

(3) **Bottleneck Projection:** Finally, a projection layer maps the context-rich features to the latent distribution, which is regularized by the STAR constraint field as defined in Eq. 5.

**Decoder.** The decoder $\mathcal{G}_\theta$ operates symmetrically to the encoder. The sampled latent $z$ is first processed by a **Mamba Backbone** to reconstruct global semantic coherence, followed by a **Convolutional Upsampling Block** to recover fine-grained waveform details from the semantic skeleton.

### 3.3. STAR-Gen: LLM-based Flow Matching for Audio Generation

Having established STAR-VAE as a robust continuous tokenizer, we now address the downstream generation task. Current audio generation methods face a fundamental trade-off: discrete autoregressive models (Kreuk et al., 2022; Copet et al., 2023) (utilizing large language model (LLM) backbones) benefit from excellent scalability and context modeling but suffer from quantization artifacts; conversely, standard diffusion models (Liu et al., 2023a; 2025d) offer high fidelity but typically lack seamless integration with LLMs. To bridge this divide, we propose **STAR-Gen**, an LLM-based Flow Matching framework that leverages the *structured* latent space of STAR-VAE to enable high-fidelity generation without vector quantization. The architecture is illustrated on the right of Figure 2.

**LLM Decoder as a Conditional Flow Predictor.** We implement STAR-Gen by adapting a **causal Transformer Decoder** to operate as a conditional velocity estimator. Unlike traditional next-token prediction, the model learns to approximate the time-dependent vector field $v_\theta(\mathbf{z}_t, t|\mathbf{c})$ that transports the noise distribution $\mathcal{N}(0, I)$ to STAR-VAE's latent data distribution. Formally, given a text condition $\mathbf{c}$ and a noisy latent state $\mathbf{z}_t$, the training objective minimizes:

$$\mathcal{L}_{\text{FM}} = \mathbb{E}_{t,\mathbf{z}_0,\mathbf{z}_1} \left[ \|v_\theta(\mathbf{z}_t, t|\mathbf{c}) - (\mathbf{z}_1 - \mathbf{z}_0)\|^2 \right] \qquad (7)$$

where $t$ follows a logit-normal distribution (Esser et al., 2024) (i.e., $\text{logit}(t) \sim \mathcal{N}(0, 1)$), $\mathbf{z}_0 \sim \mathcal{N}(0, I)$ is the noise prior, $\mathbf{z}_1$ is a latent sample from STAR-VAE's data distribution, and the interpolation path is defined as $\mathbf{z}_t = (1-t)\mathbf{z}_0 + t\mathbf{z}_1$. By treating continuous latents as sequential inputs to the decoder, we effectively cast continuous generation as a **causal sequence modeling task**. This formulation retains the scalable inductive bias of decoder-only architectures while operating strictly in the continuous domain, thereby preserving the fine-grained spectral details encoded by STAR-VAE without quantization loss.

**Hybrid Attention Mechanism.** To adapt the autoregressive architecture for non-autoregressive flow matching (Lipman et al., 2022), we employ a hybrid masking strategy: (1)

**Causal Masking for Text:** Text conditions are processed with a causal mask, respecting the sequential dependency of natural language. (2) **Bi-directional Masking for Audio:** The noisy audio latents utilize a bi-directional mask. This allows the model to attend to the global audio context simultaneously, iteratively refining the entire sequence from noise to structure, akin to bidirectional flow matching.

## 4. Experiments

### 4.1. Experimental Setup

**Datasets.** For STAR-VAE training, we use a large-scale audio dataset comprising **Freesound** (Fonseca et al., 2017), **FMA** (Defferrard et al., 2016), and **FSD50K** (Fonseca et al., 2021). We exclude recordings with native sampling rates below 44.1kHz and those exhibiting artificial high-frequency cutoffs, standardizing all audio to 44.1kHz stereo and pruning silence segments. For STAR-Gen, we train on **Wav-Caps** (Mei et al., 2024) and **AudioCaps** (Kim et al., 2019). All evaluations are conducted on **AudioCaps Test** and **Song Describer Dataset** (Manco et al., 2023).

**Evaluation Metrics.** We employ a comprehensive suite of metrics to evaluate both reconstruction fidelity and generative realism. For reconstruction, we report **Mel-Spectrogram Distance (MSD)**, **STFT Distance (STFT-D)**, and **Scale-Invariant Signal-to-Distortion Ratio (SI-SDR)** (Le Roux et al., 2019), all computed using the `auraloss` (Steinmetz & Reiss, 2020) library with default parameters. Additionally, we compute **FAD** (Kilgour et al., 2018) on reconstructed audio to verify if the VAE latent space preserves high-level semantic features. For the downstream generation, following Stable Audio Open (Evans et al., 2025), we assess quality using $FD_{openl3}$ (Cramer et al., 2019) and **KL Divergence (KL)** using PaSST model (Koutini et al., 2021). Semantic alignment is measured via **CLAP** score (Wu et al., 2023). To quantify latent space topology, we report **Latent Correlation (LC)**, computed on the latent representations as the mean off-diagonal value of the channel-wise correlation matrix.

**Implementation Details.** We train STAR-VAE on 24 NVIDIA H800 GPUs using a robust two-stage strategy. *Phase I: Standardized Pre-training.* We start with a standard isotropic KL objective ($\beta = 1e-4$) and train for approximately 150 hours. The model is optimized using AdamW with learning rates of $1e-4$ for the autoencoder and $2e-4$ for the discriminator, utilizing an inverse square root scheduler. The loss function combines a Multi-Resolution STFT loss and a patch-based adversarial hinge loss with feature matching. *Phase II: Structured Fine-tuning.* We transition to the STAR bottleneck ($\gamma = 2.0$) for an additional ~70 hours. For STAR-Gen, we train on 8 NVIDIA H800 GPUs, initializing from a pre-trained Qwen3-0.6B (Yang et al., 2025) decoder backbone and fine-tuning for LLM-based

flow matching on STAR-VAE latents using AdamW (learning rate $1e-4$) for approximately 100 hours. Additional implementation details, including full architectural specifications, training objectives, and detailed STAR-Gen training configurations, can be found in Appendix B.

### 4.2. Main Results

**Audio Reconstruction.** We conduct a comprehensive evaluation against representative continuous baselines, including the spectrogram-based Mel-VAE (Huang et al., 2023) and waveform-based models, including High-Rate AudioGen (Kreuk et al., 2022) (latent rate 100Hz) and $\epsilon$ar-VAE (Wang et al., 2025) (43Hz), and the competitive Low-Rate Stable Audio Open (Evans et al., 2025) (21.5Hz). As reported in Table 1, our results highlight three key findings: (1) In direct comparison with Stable Audio Open (SAO) at the same 21.5Hz latent rate, STAR-VAE achieves consistent improvements across all metrics on both datasets, with significant gains in semantic preservation (FAD: $3.29 \to 2.31$ on AudioCaps, $0.69 \to 0.25$ on Song Describer) and latent regularity (LC: $0.11 \to 0.08$), demonstrating that **our structured constraint preserves semantic content and improves latent regularity more effectively than the standard isotropic KL regularization**. (2) From the perspective of downstream generation, STAR-VAE's superior semantic preservation is crucial. While the 43Hz $\epsilon$ar-VAE achieves better signal-level metrics due to its lower compression rate, STAR-VAE significantly outperforms it in semantic quality (FAD: 2.31 vs. 4.44 on AudioCaps) and latent regularity (LC: 0.08 vs. 0.13), which **directly benefits downstream generative models** that rely on high-level semantic features and structured latent distributions. (3) The ablation study confirms the necessity of STAR regularization and demonstrates the Reconstruction Drift phenomenon. Removing STAR from the hybrid CNN-Mamba architecture leads to severe degradation across all metrics, with FAD increasing to 2.74 on AudioCaps and LC to 0.10, compared to 2.31 and 0.08 for STAR-VAE. While the Mamba component provides semantic benefits, under isotropic constraints, the hybrid architecture exhibits worse reconstruction quality on spectral metrics (STFT-D: 1.35 vs. 1.28, MSD: 0.93 vs. 0.89 on AudioCaps), demonstrating the Reconstruction Drift phenomenon. This validates that STAR regularization is essential for safely deploying high-capacity sequence models. Furthermore, CNN-STAR consistently outperforms CNN-VAE across all metrics (e.g., FAD: 2.65 vs. 3.36 on AudioCaps), confirming that STAR's effectiveness is architecture-agnostic and not limited to hybrid CNN-Mamba structures.

**Audio Generation.** To assess the generative capability of the latent space, we evaluate performance on **Text-to-Audio (T2A)** generation on AudioCaps. We compare **STAR-Gen** against SOTA baselines: **AudioLDM 2-large** (Liu et al.,

*Table 1.* Reconstruction fidelity comparison on AudioCaps (Sound) and Song Describer (Music). All baselines use official model weights for evaluation. Different autoencoders use various sampling rates, but evaluations are conducted at 44.1kHz for a fair comparison. Different latent rates are not strictly comparable. ↓ indicates lower is better, ↑ indicates higher is better. Best results in the Target Setting of *Low-Rate* category are highlighted in **bold**. SR: sample rate, LC: latent correlation.

| Model | SR | Latent Rate | AudioCaps (Sound) | | | | | Song Describer (Music) | | | | |
|---|---|---|---|---|---|---|---|---|---|---|---|---|
| | | | STFT-D ↓ | MSD ↓ | SI-SDR ↑ | FAD ↓ | LC ↓ | STFT-D ↓ | MSD ↓ | SI-SDR ↑ | FAD ↓ | LC ↓ |
| *Baselines (Spectrogram / High-Rate)* | | | | | | | | | | | | |
| Mel-VAE | 16kHz | 31.2Hz | 2.53 | 1.72 | -34.45 | 2.86 | 0.33 | 3.04 | 1.89 | -41.88 | 0.84 | 0.25 |
| AudioGen | 48kHz | 100Hz | 2.18 | 1.41 | -1.25 | 2.36 | 0.06 | 2.62 | 1.50 | 5.55 | 1.16 | 0.02 |
| $\epsilon$ar-VAE | 44.1kHz | 43Hz | 1.08 | 0.72 | 6.13 | 4.44 | 0.13 | 0.96 | 0.57 | 11.51 | 0.29 | 0.11 |
| *Low-Rate Continuous VAEs (Target Setting)* | | | | | | | | | | | | |
| Stable Audio Open | 44.1kHz | 21.5Hz | 1.25 | 0.86 | -0.95 | 3.29 | 0.11 | 1.59 | 0.88 | 5.78 | 0.69 | 0.09 |
| **STAR-VAE (Ours)** | 44.1kHz | 21.5Hz | **1.17** | **0.75** | **-0.03** | **2.31** | **0.08** | **1.32** | **0.80** | **6.40** | **0.25** | **0.08** |
| *Hybrid CNN-Mamba (w/o STAR)* | 44.1kHz | 21.5Hz | 1.35 | 0.93 | -1.43 | 2.74 | 0.10 | 1.57 | 0.91 | 4.20 | 0.39 | 0.10 |
| *CNN-STAR (w/o Mamba)* | 44.1kHz | 21.5Hz | 1.22 | 0.81 | -0.35 | 2.65 | 0.09 | 1.40 | 0.84 | 5.58 | 0.38 | 0.08 |
| *CNN-VAE (w/o STAR, w/o Mamba)* | 44.1kHz | 21.5Hz | 1.28 | 0.89 | -1.14 | 3.36 | 0.11 | 1.46 | 0.86 | 5.02 | 0.45 | 0.12 |

2024b), **Stable Audio Open** (Evans et al., 2025), **Tango 2** (Majumder et al., 2024), and **TangoFlux** (Hung et al., 2024) (w/o CLAP reward model for fair comparison).

*Table 2.* Generative quality comparison on Text-to-Audio generation evaluated on AudioCaps test set. All baselines use official model weights and configurations. Generated audio samples are trimmed to 10 seconds to ensure consistent evaluation.

| MODEL | PARAMS | FD$_{\text{OPENL3}}$ ↓ | KL ↓ | CLAP ↑ |
|---|---|---|---|---|
| AUDIOLDM 2-LARGE | 712M | 108.3 | 1.81 | 0.42 |
| TANGO 2 | 866M | 108.4 | 1.11 | 0.44 |
| TANGOFLUX | 515M | 80.2 | 1.22 | 0.43 |
| STABLE AUDIO OPEN (SAO) | 1.05B | 89.2 | 2.58 | 0.29 |
| *SAO w/ STAR-VAE* | 1.05B | 72.5 | 2.15 | 0.35 |
| **STAR-GEN (OURS)** | 905M | **55.8** | **1.09** | **0.48** |
| *STAR-Gen w/ SAO-VAE* | 905M | 67.4 | 1.21 | 0.44 |
| *STAR-Gen w/ $\epsilon$ar-VAE* | 905M | 76.45 | 1.53 | 0.41 |

In Table 2, results highlight three findings: (1) **STAR-Gen achieves SOTA performance on T2A generation across all metrics**, substantially outperforming all baselines in quality (FD$_{\text{openl3}}$: 55.8 vs. 80.2 for the best baseline TangoFlux), perceptual realism (KL: 1.09 vs. 1.11 for Tango 2), and semantic alignment (CLAP: 0.48 vs. 0.44 for Tango 2). (2) **STAR-VAE benefits traditional diffusion models**. Replacing SAO's VAE with STAR-VAE (SAO w/ STAR-VAE) improves performance across all metrics (FD$_{\text{openl3}}$: 89.2 → 72.5, KL: 2.58 → 2.15, CLAP: 0.29 → 0.35). (3) STAR-VAE's structured latent space also benefits our LLM-based Flow Matching model. STAR-Gen achieves superior performance (FD$_{\text{openl3}}$: 55.8, KL: 1.09, CLAP: 0.48), demonstrating **the universal effectiveness of the structured latent space for both standard diffusion models and our LLM-based continuous trajectory modeling approach**.

### 4.3. Ablation Studies and Analysis

**1. Impact of Latent Topology on Information Packing.** To decode the internal organization of the latent space, we analyze the distribution of information and its contribution to reconstruction fidelity. (1) **Channel-wise Information Distribution.** Figure 3(a) visualizes the channel-wise KL divergence. The Isotropic Baseline exhibits chaotic, multimodal distribution with "Disordered Outliers" (e.g., indices 33, 53), confirming **Disordered Information Packing** (Sec. 2): uniform KL penalty lacks inductive bias, fragmenting information arbitrarily. In contrast, STAR-VAE shows structured, monotonically decreasing distribution aligned with **Capacity Gradient** (Sec. 3.1): low-index channels (High-Capacity Zone) store structural features, high-index channels (Low-Capacity Zone) store stochastic residuals, confirming Gamma-Growth penalty induces hierarchical organization. (2) **Information Hierarchy Validation.** We validate hierarchy via *inference-time channel truncation* analysis (Figure 3(b)) by progressively reconstructing audio using only the top-$k$% channels. STAR-VAE exhibits "PCA-like Energy Compaction": reconstruction error drops precipitously, converging to near-optimal fidelity with top 37.5% channels, indicating that head channels capture the majority of structural information. Conversely, the Baseline suffers from Information Dilution—error remains high even up to the top 90% channels, demonstrating inefficient information scattering that creates a noisy optimization landscape for downstream generation.

**2. Spectral Fidelity Across Frequency Bands.** We analyze the spectral error distribution across the full bandwidth in Figure 3(c). A critical divergence emerges in the high-frequency regime ($> 18$kHz). The Isotropic Baseline shows a **sharp surge** in distortion, with STFT-distance increasing from 1.4 at 14kHz to 2.3 at 22kHz, indicating a failure to model fine-grained spectral details. In contrast, STAR-VAE maintains a **flattened trajectory** in this region, with STFT-distance increasing only from 1.2 to 1.8 over the same frequency range. This demonstrates that STAR's inductive ordering, which routes high-entropy stochastic details to high-index channels, enables effective learning of high-frequency textures, ensuring consistent reconstruction fidelity across the full frequency spectrum. Additional visual comparisons of spectrograms are in Appendix D.

**3. Architecture: Why Mamba?** To validate the ar-

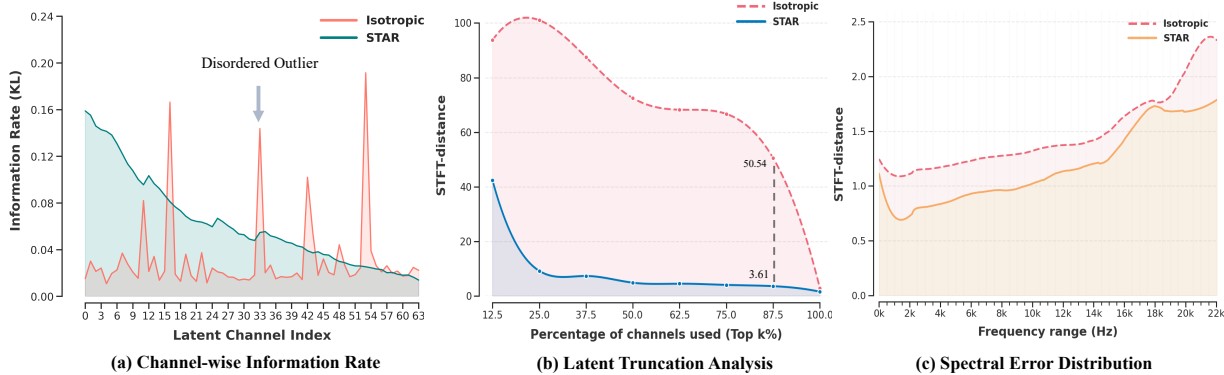

(a) Channel-wise Information Rate     (b) Latent Truncation Analysis     (c) Spectral Error Distribution

*Figure 3.* Quantitative comparison of latent topology and spectral fidelity between isotropic KL regularization and STAR regularization. **(a) Channel-wise Information Rate:** Average KL divergence plotted against latent channel index. **(b) Latent Truncation Analysis:** Reconstruction error (STFT-distance) as a function of the percentage of top-ranked channels retained. **(c) Spectral Error Distribution:** Reconstruction error magnitude across the full frequency spectrum (0–22kHz).

chitectural choice, we ablate the Mamba component in STAR-VAE's hybrid architecture. Table 3 shows that CNN-STAR exhibits the worst reconstruction quality, confirming that limited receptive fields harm both spectral fidelity and global semantic structure. While Transformer-STAR matches Mamba's capacity for global dependencies, it incurs a higher computational cost due to quadratic attention scaling. Mamba-STAR achieves the best overall performance, offering superior semantic consistency and fidelity while maintaining faster inference speeds, confirming Mamba as the optimal backbone balancing global context modeling with linear computational efficiency.

*Table 3.* Architecture ablation on Song Describer Dataset. All models use STAR regularization. CNN-STAR: w/o Mamba; Transformer-STAR: Mamba→Transformer.

| ARCHITECTURE | STFT-D ↓ | MSD ↓ | SI-SDR ↑ | FAD ↓ | INFER TIME (S) ↓ |
|---|---|---|---|---|---|
| CNN-STAR | 1.40 | 0.84 | 5.58 | 0.38 | **0.68** |
| TRANSFORMER-STAR | 1.35 | 0.81 | 5.96 | 0.30 | 0.92 |
| **MAMBA-STAR (STAR-VAE)** | **1.32** | **0.80** | **6.40** | **0.25** | 0.85 |

**4. More Findings.** We present additional quantitative ablation results in Appendix C. (1) **Growth Function Ablation:** The power-law Gamma-Growth function with $\gamma = 2.0$ outperforms Step and Linear functions, validating that convex allocation ($\gamma > 1$) widens capacity for structural components. (2) **STAR-Gen Scalability:** Performance improves with model scale, confirming STAR-Gen benefits from increased LLM capacity. (3) **CFG and Inference Analysis:** The optimal balance is achieved with CFG scale 3.0 and inference steps 24. (4) **Linear Probing:** STAR-VAE preserves more semantic information than isotropic baselines on classification and retrieval tasks.

## 5. Related Work

**VAEs for Audio Synthesis.** The efficacy of Latent Diffusion Models (LDMs) (Rombach et al., 2022) and Flow Matching systems (Lipman et al., 2022) relies heavily on the quality of the continuous latent representations learned by VAEs (Pin-

heiro Cinelli et al., 2021). Early pivotal works, such as **AudioGen** (Kreuk et al., 2022), utilized discrete quantization (VQ-VAE) to compress audio, while **AudioLDM** (Liu et al., 2023a) demonstrated the effectiveness of continuous VAEs operating on Mel-spectrograms for text-to-audio generation. More recently, **Stable Audio Open** (Evans et al., 2025) pushed the boundaries of high-fidelity reconstruction by scaling convolutional autoencoders with advanced discriminators, enabling the generation of stereo audio with varying durations. While recent works like $\epsilon$**ar-VAE** (Wang et al., 2025) have begun exploring sequence-aware architectures to address this issue, they still largely rely on isotropic priors, failing to explicitly model the hierarchical topology of audio. Our STAR-VAE bridges this gap by synergizing the global context modeling of Mamba with a topology-aware regularization strategy, ensuring both high-fidelity reconstruction and a structured latent space optimized for generation. Appendix A details related general audio generation models and the differentiators of our STAR-Gen.

## 6. Conclusion

In this work, we formalized the *Rate-Distortion-Regularity Trilemma* in continuous audio VAEs, identifying the isotropic Gaussian prior as the root cause of disordered information packing. To resolve this, we proposed **Structured Topology-Aware Regularization (STAR)**, a general training strategy that reshapes latent space geometry by imposing a capacity gradient aligned with audio's hierarchical structure. We introduced **STAR-VAE**, combining STAR with a hybrid CNN-Mamba architecture for SOTA reconstruction, and **STAR-Gen**, an LLM-based Flow Matching framework leveraging STAR-VAE's structured latent space for high-fidelity generation. Extensive experiments demonstrate that STAR resolves the trilemma across diverse architectures and domains, establishing a superior continuous tokenization paradigm for neural audio generation.

## Impact Statement

This paper presents work whose goal is to advance the field of machine learning, specifically in the domain of continuous representation learning for **sound effect and music synthesis**. Our proposed STAR-VAE framework improves the fidelity and efficiency of audio reconstruction, with potential applications in creative content production (e.g., sound design, music composition) and high-efficiency neural audio compression.

Since our work focuses on modeling general acoustic structures rather than speech synthesis or voice cloning, the risks associated with identity impersonation are minimal. However, we acknowledge that high-fidelity generative audio models raise broader questions regarding intellectual property and the potential creation of misleading acoustic environments (e.g., synthetic sound effects for misinformation). We encourage the research community to continue developing robust watermarking and attribution techniques to ensure the responsible deployment of generative audio technologies.

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

# A. Related Work: General Audio Generation

The field of audio generation has witnessed a rapid paradigm shift from discrete autoregressive modeling to continuous latent generative frameworks. Pioneered by **AudioGen** (Kreuk et al., 2022) and **MusicGen** (Copet et al., 2023), early approaches treated audio synthesis as a sequence modeling task over discrete tokens derived from VQ-VAEs (Van Den Oord et al., 2017). While effective, these methods often face a trade-off between quantization artifacts and excessive sequence lengths. Consequently, the paradigm has shifted towards continuous latent spaces, where Latent Diffusion Models (LDMs)—exemplified by **AudioLDM** (Liu et al., 2023a)—achieve high-fidelity synthesis by modeling the probability density of continuous audio latents. Recently, this has been further advanced by Flow Matching and Diffusion Transformers (Liu et al., 2023b; Shi et al., 2025; Liu et al., 2025c). Notably, **TangoFlux** (Hung et al., 2024) demonstrates the efficiency of flow matching for fast inference. In parallel, **ThinkSound** (Liu et al., 2025a) significantly improves semantic consistency, while **Stable Audio Open** (Evans et al., 2025) extends capabilities to high-fidelity long-form audio generation. However, despite the diversity of these generative backbones, their ultimate fidelity is strictly upper-bounded by the representational capacity of the underlying Variational Autoencoder. STAR-VAE addresses this foundational bottleneck, proposing a topologically structured VAE that serves as a superior continuous tokenizer for these advanced generative priors. Building upon STAR-VAE's structured latent space, we introduce **STAR-Gen**, an LLM-based Flow Matching framework that bridges discrete autoregressive modeling and continuous generation, enabling high-fidelity audio synthesis without quantization artifacts while retaining the scalability and context modeling advantages of large language models.

# B. Implementation Details

## B.1. Architecture Configuration

Our STAR-VAE builds upon the convolutional backbone of Stable Audio Open (Evans et al., 2025), augmented with bidirectional Mamba adapters for global context modeling.

**Convolutional Backbone.** The encoder processes raw waveforms through 5 convolutional blocks. Each block performs downsampling and channel expansion via strided convolutions. Before each downsampling operation, we employ a series of ResNet-like (He et al., 2016) layers utilizing dilated convolutions and Snake activations (Ziyin et al., 2020) to capture multi-scale temporal features. The decoder mirrors this structure, using transposed strided convolutions for upsampling with channel contraction.

**Mamba Integration.** We integrate bidirectional Mamba (Gu & Dao, 2024) blocks into the bottleneck of both the encoder and decoder. Specifically, we insert a sequence of 2 Mamba layers ($d_{state} = 16, d_{conv} = 4$, expansion=2) before the final encoder convolution and after the first decoder upsampling. This allows the model to capture long-range dependencies that exceed the receptive field of the convolutional stack.

**Normalization Strategy.** We empirically found that standard normalization techniques failed to stabilize training. This instability arises from the interaction between the unbounded nature of Snake activations and the recurrent dynamics of Mamba, which can lead to rapid variance explosion in the hidden states. To mitigate this, we adopt a strict *Pre-Norm* strategy with Layer Normalization applied before every Mamba block and Residual connection, ensuring bounded signal propagation throughout the deep network.

## B.2. Training Objectives

**Reconstruction Loss ($\mathcal{L}_{\textbf{Rec}}$).** We employ a Multi-Resolution STFT loss with A-weighting (Fletcher & Munson, 1933) to match human perception. Window lengths are set to $\{2048, 1024, 512, 256, 128, 64, 32\}$.

**Adversarial Loss ($\mathcal{L}_{\textbf{Adv}}$).** We utilize a patch-based hinge loss with a Multi-Scale STFT Discriminator ensemble (window lengths $\{2048, \ldots, 128\}$). The adversarial loss combines both adversarial and feature matching objectives, where feature matching minimizes the $L_1$ distance between discriminator feature maps of real and reconstructed audio.

**Regularization ($\mathcal{L}_{\textbf{KL}}$).** In Phase I, this is the standard isotropic KL divergence as defined in Eq. 3. In Phase II, it transitions to the STAR objective (Eq. 5) with $\gamma = 2.0$, enforcing the structured capacity gradient.

The total training objective $\mathcal{L}_{\text{Total}}$ follows Eq. 1:

$$\mathcal{L}_{\text{Total}} = \mathcal{L}_{\text{Rec}}(x, \hat{x}) + \lambda_{\text{Adv}}\mathcal{L}_{\text{Adv}}(x, \hat{x}) + \beta\mathcal{L}_{\text{KL}}(q_\phi || p) \tag{8}$$

where $\lambda_{\text{Adv}} = 0.1$. The regularization term $\mathcal{L}_{\text{KL}}$ varies by phase: in Phase I, $\beta = 1e-4$ for uniform KL divergence; in Phase II, it transitions to the STAR objective, where the channel-wise weights $\beta_c$ follow the Gamma-Growth function (Eq. 4) with $\beta_{\max} = 4e-4$.

## B.3. STAR-Gen Training Details

We train STAR-Gen on 8 NVIDIA H800 GPUs using the WavCaps and AudioCaps datasets. The model is initialized from a pre-trained Qwen3-0.6B decoder backbone and fine-tuned for flow matching on STAR-VAE latents. We use the AdamW optimizer with a learning rate of $1e-4$, a constant learning-rate schedule with 2,000 warmup steps, and an exponential moving average (EMA) of model parameters with decay 0.9999. The training objective follows the Flow Matching formulation in Section 3.3, where $t$ is sampled from a logit-normal distribution. We train for approximately 100 hours until convergence. For efficiency, we pack multiple training examples into a single long sequence per iteration, with the total input length capped at 8192 tokens.

# C. Additional Quantitative Results

## C.1. Comparison of Growth Functions and Gamma Hyperparameter Ablation

As discussed in Section 3.1, we theoretically advocate for a convex allocation ($\gamma > 1$) to match audio's heavy-tailed spectral distribution. To validate this design choice, we conduct ablation studies comparing different growth functions (Step, Linear, and Gamma-Growth) and various $\gamma$ values. Table 4 presents reconstruction quality on AudioCaps across different growth functions and $\gamma$ parameters.

The results validate our theoretical analysis across three key dimensions: (1) **Step Function vs. Continuous Gradients:** The Step function, which imposes a hard binary threshold, achieves the worst performance across all metrics (STFT-D: 1.58, FAD: 3.15, LC: 0.11), demonstrating that the abrupt transition creates spectral discontinuities and fails to preserve high-frequency details. This validates the necessity of a continuous gradient design. (2) **Concave vs. Convex Allocation:** The concave Gamma-Growth ($\gamma = 0.5$) underperforms Linear ($\gamma = 1.0$) in reconstruction quality (STFT-D: 1.42 vs. 1.38, FAD: 2.75 vs. 2.65), as the rapid penalty increase in low-index channels prematurely compresses the Structure Subspace, forcing the model to discard critical semantic information. While the concave allocation achieves lower LC (0.07), this comes at the cost of degraded reconstruction fidelity, confirming that concave allocation ($\gamma < 1$) is suboptimal. (3) **Optimal Convexity:** Gamma-Growth with $\gamma = 2.0$ achieves the best performance across all metrics (STFT-D: 1.17, FAD: 2.31, LC: 0.08), effectively widening the "Safe Harbor" for structural components while maintaining balanced latent representation. When $\gamma$ increases to 3.0, the over-convex allocation leads to performance degradation (STFT-D: 1.25, FAD: 2.52, LC: 0.09), as the penalty becomes too lenient for most channels, reducing the model's incentive for efficient information packing.

*Table 4.* Comparison of growth functions and $\gamma$ hyperparameter ablation based on reconstruction performance on AudioCaps. All models use the same hybrid CNN-Mamba architecture with STAR regularization.

| Growth Function | $\gamma$ | STFT-D $\downarrow$ | MSD $\downarrow$ | FAD $\downarrow$ | LC $\downarrow$ |
|---|---|---|---|---|---|
| Step | - | 1.58 | 0.95 | 3.15 | 0.11 |
| Linear | 1.0 | 1.38 | 0.88 | 2.65 | 0.08 |
| Gamma-Growth | 0.5 | 1.42 | 0.90 | 2.75 | **0.07** |
| Gamma-Growth | 1.5 | 1.32 | 0.82 | 2.45 | 0.08 |
| Gamma-Growth | **2.0** | **1.17** | **0.75** | **2.31** | 0.08 |
| Gamma-Growth | 3.0 | 1.25 | 0.78 | 2.52 | 0.09 |

## C.2. STAR-Gen Scalability Analysis

To investigate the scalability of STAR-Gen, we evaluate performance across two LLM base model sizes: Qwen3-0.6B and Qwen3-1.7B. Table 5 reports generation quality on the AudioCaps test set. Results demonstrate improvements with LLM scale: the Qwen3-1.7B backbone outperforms the Qwen3-0.6B backbone in quality ($FD_{\text{openl3}}$: 54.1 vs. 55.8) and semantic alignment (CLAP: 0.51 vs. 0.48), while maintaining similar perceptual realism (KL: 1.05), confirming that STAR-Gen benefits from increased LLM capacity while maintaining efficiency through the con-

*Table 5.* Scalability analysis of STAR-Gen across different LLM base model sizes on AudioCaps test set. All models are trained with the same configuration and evaluated using the same metrics. Model sizes refer to the underlying Qwen3 LLM backbone.

| LLM Size | $FD_{\text{openl3}} \downarrow$ | KL $\downarrow$ | CLAP $\uparrow$ |
|---|---|---|---|
| Qwen3-0.6B | 55.8 | 1.09 | 0.48 |
| **Qwen3-1.7B** | **54.1** | **1.05** | **0.51** |

tinuous latent representation.

## C.3. CFG Scale and Inference Step Analysis

We analyze the impact of Classifier-Free Guidance (CFG) scale (Ho & Salimans, 2022) and inference steps on STAR-Gen's generation quality. Figure 4 presents the performance landscape across CFG scales ranging from 2.0 to 7.0 and inference steps from 10 to 50. Our analysis reveals three key observations: (1) **CFG Scale Optimization:** As shown in Figure 4(a), $FD_{openl3}$ exhibits a U-shaped relationship with CFG scale, with optimal performance achieved in the range of 3.0–5.0. Values below this range result in insufficient conditioning, while higher values lead to over-conditioning and quality degradation. (2) **Semantic Alignment:** Figure 4(b) demonstrates that CLAP scores peak within the same CFG range (3.0–5.0), confirming that moderate CFG values enhance audio–text alignment without compromising perceptual quality. (3) **Inference Efficiency:** Figure 4(c) shows that $FD_{openl3}$ improves monotonically with inference steps, but the marginal gains diminish significantly beyond 20–30 steps, indicating diminishing returns for additional computational cost. Based on these findings, we adopt CFG scale 3.0 and inference steps 24 as our final configuration, providing an optimal balance between generation quality and computational efficiency.

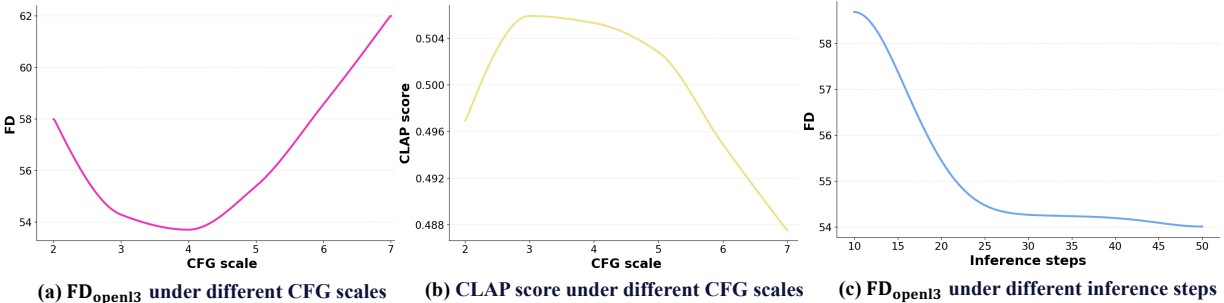

(a) $FD_{openl3}$ **under different CFG scales**  (b) **CLAP score under different CFG scales**  (c) $FD_{openl3}$ **under different inference steps**

*Figure 4.* Performance landscape of STAR-Gen under different CFG scales and inference steps on the AudioCaps test set.

## C.4. Linear Probing for Semantic Information

To evaluate the semantic information preserved in STAR-VAE's latent space, we conduct linear probing experiments using Support Vector Machines (SVM) on a binary audio classification task. We compare STAR-VAE against a CNN-VAE baseline (without Mamba and STAR components) trained with identical data and training duration to ensure fair comparison. Both models extract frozen latent representations, on which we train SVM classifiers to distinguish between two semantic categories: human activities and natural sounds. Table 6 reports the classification accuracy. Results demonstrate that STAR-VAE achieves higher classification accuracy (70.32%) compared to the CNN-VAE baseline (65.02%), indicating that the structured latent space induced by STAR regularization preserves more semantic information, validating that the hierarchical capacity allocation effectively captures high-level audio semantics.

*Table 6.* Linear probing results comparing semantic information preservation in STAR-VAE and CNN-VAE baseline.

| Model | Classification Acc. ↑ |
|---|---|
| CNN-VAE (Baseline) | 65.02 |
| **STAR-VAE (Ours)** | **70.32** |

## C.5. Human Evaluation (Mean Opinion Score)

To complement the objective metrics reported in the main paper, we conduct subjective Mean Opinion Score (MOS) studies on both reconstruction and generation, following standard listening-test protocol. For each task, 15 listeners with normal hearing rated randomly sampled audio clips on a 1–5 scale (1: bad, 5: excellent) for overall perceptual quality. Clips were presented in randomized order with anonymized model identities to mitigate rater bias, and we report mean ± std across listeners and clips.

**Reconstruction MOS.** We evaluate 60 clips drawn from two domains: 30 general-audio clips from the AudioCaps test set (Kim et al., 2019) and 30 music clips from the Song Describer Dataset (Manco et al., 2023), enabling a domain-balanced assessment. We compare STAR-VAE against the music-oriented sequence-aware baseline $\epsilon$ar-VAE (Wang et al., 2025) and the high-fidelity convolutional baseline Stable Audio Open (SAO) (Evans et al., 2025), along with Ground Truth as the

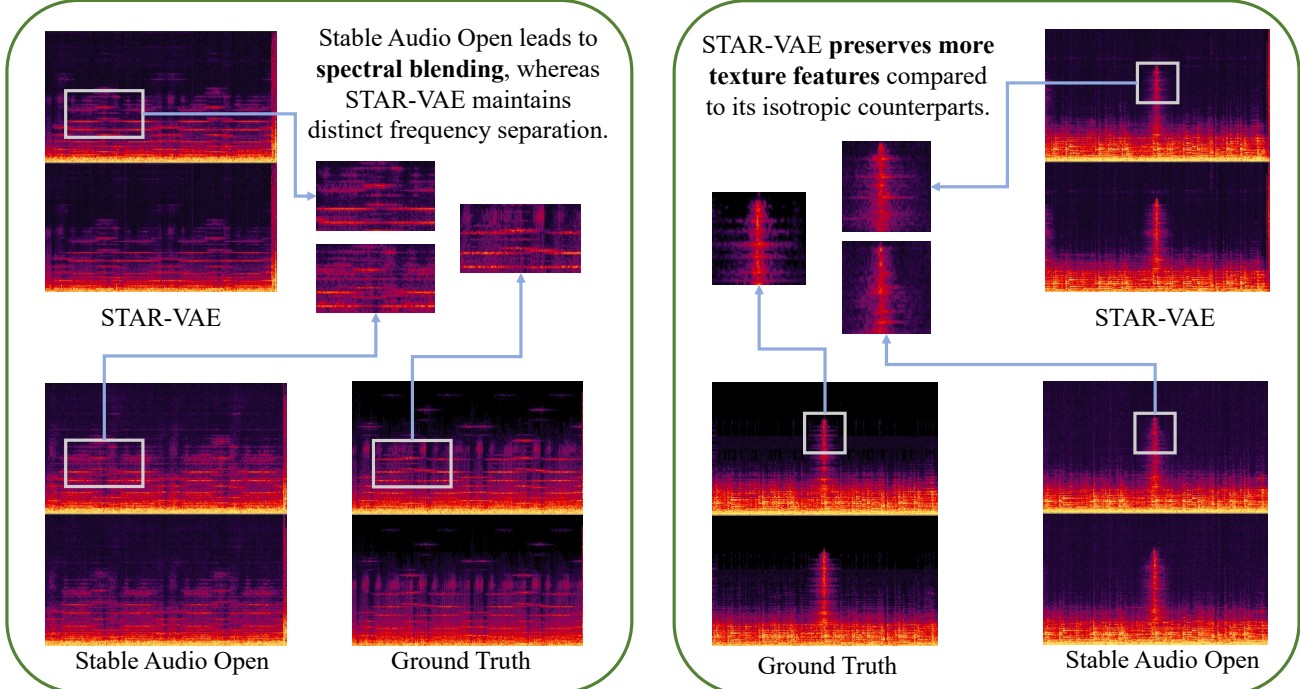

*Figure 5.* Visual comparison of audio reconstruction between STAR-VAE and Stable Audio Open.

upper bound. Crucially, STAR-VAE and SAO operate at a matched latent rate of 21.5 Hz, isolating the contribution of our topology-aware design from compression-ratio confounds, whereas $\epsilon$ar-VAE uses a higher 43 Hz latent rate. As shown in Table 7, STAR-VAE achieves $4.32 \pm 0.12$ MOS, narrowing the gap to Ground Truth ($4.45 \pm 0.10$) to within 0.13 points while outperforming SAO by 0.27 points at the same latent rate, and surpassing $\epsilon$ar-VAE despite using half the latent rate.

*Table 7.* Reconstruction MOS on 60 clips (30 AudioCaps + 30 Song Describer Dataset), rated by 15 listeners on a 1–5 scale. STAR-VAE and SAO share the same 21.5 Hz latent rate.

| Model | Latent Rate (Hz) | MOS ↑ |
|---|---|---|
| Ground Truth | – | $4.45 \pm 0.10$ |
| **STAR-VAE (Ours)** | 21.5 | **$4.32 \pm 0.12$** |
| $\epsilon$ar-VAE (Wang et al., 2025) | 43 | $4.21 \pm 0.14$ |
| SAO (Evans et al., 2025) | 21.5 | $4.05 \pm 0.15$ |

**Generation MOS.** For text-to-audio generation, we evaluate 60 clips synthesized from prompts in the AudioCaps test set (Kim et al., 2019), comparing STAR-Gen against two strong baselines: TangoFlux (Hung et al., 2024) and SAO (Evans et al., 2025), with Ground Truth recordings included as a reference. Table 8 shows that STAR-Gen attains the highest MOS among all baselines ($3.92 \pm 0.16$), outperforming TangoFlux ($3.76 \pm 0.19$) and SAO ($3.71 \pm 0.18$). These subjective results corroborate our quantitative findings: STAR-Gen's improved generation quality stems from operating on the structured latent space of STAR-VAE, where the topology-aware regularization provides a smoother manifold for downstream Flow Matching, translating into perceptually more natural audio.

*Table 8.* Generation MOS on 60 text-to-audio clips synthesized from AudioCaps prompts, rated by 15 listeners on a 1–5 scale.

| Model | MOS ↑ |
|---|---|
| Ground Truth | $4.25 \pm 0.12$ |
| **STAR-Gen (Ours)** | **$3.92 \pm 0.16$** |
| TangoFlux (Hung et al., 2024) | $3.76 \pm 0.19$ |
| SAO (Evans et al., 2025) | $3.71 \pm 0.18$ |

Taken together, these subjective evaluations confirm that the gains observed under objective metrics translate to perceivable improvements for human listeners, both in reconstruction fidelity at matched latent rate and in end-to-end generation quality.

## D. Qualitative Analysis

We provide a visual assessment of reconstruction fidelity by comparing STAR-VAE against the state-of-the-art baseline, *Stable Audio Open*. Figure 5 illustrates the structural advantages of our anisotropic prior across diverse audio samples.

As highlighted in the comparison, isotropic models often suffer from **spectral blending**, where distinct frequency bands merge due to over-smoothing (Figure 5, left). In contrast, STAR-VAE maintains distinct frequency separation, successfully preventing the "isotropic blurring" artifacts. Furthermore, regarding transient dynamics, STAR-VAE demonstrates superior preservation of **fine-grained texture features** (Figure 5, right), whereas the baseline tends to smear these sharp spectral details. This visual evidence confirms that STAR's hierarchical capacity allocation effectively protects deterministic structural information from being washed out by the noise floor.

## E. Limitations and Future Work

In this work, we introduced **STAR-VAE** and **STAR-Gen**, establishing a novel paradigm for continuous audio tokenization and generation. By rigorously aligning the latent topology with the spectral hierarchy of audio, our method achieves state-of-the-art reconstruction and generation fidelity without the quantization artifacts inherent to discrete codecs. While our current framework sets a strong baseline, several promising directions remain for future exploration:

- First, we currently employ a static, theoretically grounded Gamma-Growth function to enforce the information hierarchy, which ensures training stability and interpretability. A natural evolution of this work is to explore content-adaptive topology, where the capacity curve $\beta$ is dynamically predicted from the input signal. This would allow the model to optimally reallocate bandwidth in real-time for highly non-stationary audio segments, further pushing the compression-fidelity frontier.

- Second, regarding sequence modeling, we leveraged the Mamba backbone to achieve linear memory scaling $\mathcal{O}(L)$, enabling the processing of long-form audio that is computationally prohibitive for Transformers. While our implementation validates this efficiency, the hardware-aware optimization of State Space Models is an active area of research. As the ecosystem matures, we expect the wall-clock training efficiency of STAR-VAE to improve further, bridging the gap between theoretical complexity and empirical speed.

- Finally, the core principle of STAR—structuring latent space by information density—is fundamentally modality-agnostic. We envision extending this continuous tokenization philosophy to unified audio understanding and generation, potentially offering a unified, scalable alternative to vector quantization in next-generation multimodal foundation models.

