# OpenReview forum: "STAR-VAE: Structured Topology-Aware Regularization for Audio Reconstruction and Generation"
_ICML.cc/2026/Conference — ICML 2026 regular_

### Official Review · Reviewer_3v9B · 2026-03-10

**Soundness:** 4
**Presentation:** 3
**Significance:** 3
**Originality:** 3
**Overall Recommendation:** 6
**Confidence:** 3

**Summary:**

The paper identifies that the trilemma in continuous VAEs for audio generation stems from a critical **topological mismatch**. Specifically, the commonly used isotropic Gaussian prior imposes a flat latent geometry that fails to capture the hierarchical structure of audio signals, where low-frequency components are structured and compressible, while high-frequency components are stochastic and largely incompressible. This mismatch leads to disordered information packing, where essential semantic features become randomly interleaved with high-entropy noise.

To address this issue, the authors propose **Structured Topology-Aware Regularization (STAR)**, a general training strategy that reshapes the latent space geometry by introducing a growth-based constraint field, which routes structural and textural information into channel subspaces with appropriate capacities.

Building on this idea, the paper further presents **STAR-VAE** and **STAR-Gen**, which achieve state-of-the-art performance on several audio reconstruction and generation tasks.

**Compliance With Llm Reviewing Policy:**

Affirmed.

**Final Justification:**

This work aligns with my initial judgment, so I will keep the score unchanged.

**Key Questions For Authors:**

The paper proposes the STAR framework for continuous VAEs in audio modeling by leveraging the intrinsic characteristics of audio signals, and demonstrates substantial improvements in reconstruction quality.

I am curious whether the core idea of STAR could be extended to **discrete VAE frameworks**. In particular, what would be the potential benefits, limitations, or practical challenges when applying similar topology-aware regularization to discrete latent representations?

Intuitively, the issues discussed in the paper—such as mismatched latent structure and disordered information allocation—may also arise in discrete VAE settings. If STAR could be adapted to discrete VAEs, it might potentially benefit **autoregressive (AR) audio generation pipelines** that rely on discrete token representations. It would be interesting if the authors could comment on this possibility.

**Limitations:**

Yes

**Strengths And Weaknesses:**

The paper provides extensive experimental results and targeted analyses. STAR-VAE shows significant improvements over strong baselines on audio reconstruction tasks on the AudioCaps and SongDescriber datasets. In addition, STAR-Gen achieves clear gains over baseline methods in text-to-audio generation.
Section 2 presents a clear and detailed explanation of the *Isotropic Prior Hypothesis*, which helps readers better understand the motivation and theoretical basis of the proposed approach.
STAR-VAE addresses an important yet largely overlooked issue in audio generation, substantially improving both reconstruction fidelity and generation quality.

---

> ### Author Rebuttal · Authors · 2026-03-30
>
> We are sincerely grateful for the reviewer's overall recommendation of **Strong Accept** (Soundness: **Excellent**) and for highlighting that STAR addresses *"an important yet largely overlooked issue"* achieving *"state-of-the-art performance".* We hope our response below could fully address your interesting question.
>
> > Q: Can STAR's core idea extend to discrete neural codecs?
>
> This is an excellent question that touches on the fundamental difference between continuous audio representation and discrete audio tokenization paradigms.
>
> **Why STAR cannot be directly applied to discrete codecs:** STAR operates on continuous VAEs where the KL bottleneck regularizes the latent distribution toward a smooth, continuous manifold. This design serves downstream diffusion/flow-based generators that require stable training on continuous distributions. In contrast, discrete neural codecs (e.g., EnCodec, DAC) output discrete tokens via codebook quantization—there is *no explicit KL term* to reshape, making STAR's channel-wise KL weighting inapplicable *in its current form*.
>
> **What insights of STAR remain transferable:** The core motivation behind STAR—that audio signals exhibit a hierarchical information structure where global content (pitch, rhythm) is structured while local detail (transients, noise) is stochastic (Section 3.1)—is independent of continuous vs. discrete representation paradigms. For discrete codecs serving AR LLM-based generation, this insight suggests that designing tokenizers with explicit awareness of this hierarchy (e.g., ensuring coarse semantic information is captured in early quantization stages) could improve AR modeling efficiency. However, the specific implementation would require rethinking quantization strategies rather than adapting KL weighting, and we leave concrete exploration to future work.

---

> > ### Author Rebuttal · Reviewer_3v9B · 2026-04-03
> >
> > The related rebuttal has been reviewed. After also considering the comments from the other reviewers, I will keep my score unchanged.

---

> > > ### Author Response · Authors · 2026-04-03
> > >
> > > We sincerely appreciate your continued strong support and are pleased to know that our rebuttal has fully addressed your concerns. Thank you again for your careful review and constructive feedback.

---

### Official Review · Reviewer_Tf89 · 2026-03-10

**Soundness:** 3
**Presentation:** 3
**Significance:** 3
**Originality:** 3
**Overall Recommendation:** 6
**Confidence:** 4

**Summary:**

This paper addresses the topological mismatch in continuous VAEs for audio, which the authors formalize as the "Rate-Distortion-Regularity Trilemma". To solve this, the paper proposes Structured Topology-Aware Regularization (STAR), a formulation that imposes a hierarchical capacity gradient across latent channels rather than relying on the standard uniform isotropic Gaussian prior. Building on this, the authors introduce STAR-VAE, and STAR-Gen, an LLM-based flow-matching model for audio generation. Quantitative experiments demonstrate STAR-Gen outperforms the Stable Audio Open (SAO) framework even when retrained with inputs from STAR-VAE, while comprising a lower parameter count.

**Compliance With Llm Reviewing Policy:**

Affirmed.

**Final Justification:**

The significance of this work is clearly demonstrated by the strong empirical performance of both STAR-VAE and the LLM-based STAR-Gen, which achieve impressive downstream generative results while maintaining a lower parameter count than the Stable Audio Open (SAO) framework. Across the dimensions of clarity, originality, and soundness, the initial submission was already strong, utilizing well-justified architectural decisions, such as the hybrid CNN-Mamba backbone, and thorough ablation studies.

The authors provided a highly thorough and convincing rebuttal that fully resolved my initial concerns, prompting me to increase my overall recommendation to a Strong Accept. By supplying the matched 43 Hz baseline, the authors successfully isolated the benefits of the STAR formulation from the compression bottleneck, empirically proving that STAR-VAE outperforms ar-VAE independent of the latent rate. Furthermore, the inclusion of a formal MOS user study adds essential subjective validation for both reconstruction and generation, making the evaluation remarkably comprehensive and reinforcing my prior positive assessment. Finally, the authors presented a logically sound explanation for the artifact differences between direct deterministic reconstruction and probabilistic flow-matching generation. Because the rebuttal meticulously addressed all of my critiques and strengthened the paper's empirical standing, I am highly confident in recommending this work for acceptance.

**Key Questions For Authors:**

1. Would it be technically feasible to train the $\epsilon$ar-VAE architecture with the STAR formulation to provide a strictly controlled comparison at the 43 Hz latent rate? Or is there a specific technical reason this would be unfeasible?
2. At the provided supplementary audio webpage, the direct audio reconstructions from STAR-VAE present audible artifacts (likely due to the high compression of the 21.5 Hz rate) that are seemingly absent in the audio generated downstream by STAR-Gen. Do you have a hypothesis for why this happens? Is the generative model trained with some regularization objectives that were omitted from Section 3.3?

**Limitations:**

Yes

**Strengths And Weaknesses:**

**Strengths:**

1. The theoretical reasoning behind STAR is sound, and the empirical evidence presented in the channel-wise information rate and truncation analyses strongly supports the authors' claims.
2. The architectural decisions, such as the use of the hybrid CNN-Mamba backbone, are well-justified via thorough ablation studies.
3. Replacing SAO’s standard embeddings with those of STAR-VAE substantially increases SAO's downstream generative performance across all metrics (Table 2). This is a compelling result that proves the quality of the STAR-VAE latent space.

**Weaknesses:**

1. While STAR-VAE uses a 21.5 Hz latent compression rate, its main competitor in the waveform domain, $\epsilon$ar-VAE, operates at a 43 Hz latent rate. As the authors note, this gives $\epsilon$ar-VAE an inherent advantage in signal-level reconstruction metrics (e.g., STFT-D, SI-SDR). Because of this mismatch, it is difficult to fully isolate the benefits of the STAR formulation from the effects of the compression bottleneck. A correct comparison (e.g., applying STAR to $\epsilon$ar-VAE or training a 43 Hz version of STAR-VAE) is currently missing.
2. This study relies entirely on objective metrics (FAD, FDopenl3, STFT-D, CLAP), it would be significantly more complete and convincing with a formal user study.

---

> ### Author Rebuttal · Authors · 2026-03-30
>
> We sincerely appreciate the reviewer's overall recommendation of **Accept** and the recognition that STAR's *"theoretical reasoning is sound"* with *"compelling"* and *"strong"* empirical support. We hope the following point-by-point response could resolve all your concerns and questions.
>
> > W1: STAR-VAE's latent rate mismatch with $\epsilon$ar-VAE (21.5 Hz vs. 43 Hz)
>
> We first clarify the design rationale: **our goal differs fundamentally from $\epsilon$ar-VAE**. STAR-VAE targets both reconstruction *and*, more importantly, downstream generation (STAR-Gen) *across diverse audio domains*, whereas $\epsilon$ar-VAE focuses on *music* reconstruction with emphasis on phase accuracy and stereophonic spatial representation. This difference motivates our choice of 21.5 Hz compression—prioritizing latent space quality for flow-matching priors over raw reconstruction fidelity.
>
> Importantly, **reconstruction metrics alone do not reflect downstream generation quality**. As shown in Table 2, despite $\epsilon$ar-VAE's reconstruction advantage (Table 1), STAR-Gen trained with $\epsilon$ar-VAE embeddings notably underperforms STAR-Gen with STAR-VAE embeddings on all generation metrics (FDopenl3, CLAP, FAD), validating our design choice.
>
> That said, for a more complete comparison isolating STAR's contribution at matched compression ratios, we compare STAR-VAE with **43 Hz latent rate (same as $\epsilon$ar-VAE)** and with 21.5 Hz on reconstruction on AudioCaps.
>
> |Model|Latent Rate|STFT-D↓|MSD↓|FAD↓|
> |-|-|-|-|-|
> |$\epsilon$ar-VAE|43 Hz|1.08|0.72|4.44|
> |STAR-VAE|43 Hz|**1.01**|**0.63**|**1.56**|
> |STAR-VAE|21.5 Hz|1.17|0.75|2.31|
>
> **Findings:** At the matched 43 Hz rate, STAR-VAE outperforms $\epsilon$ar-VAE across all metrics, confirming STAR's effectiveness independent of compression ratio.
>
> > W2: Human evaluation / user study
>
> Thank you for the suggestion. We conducted MOS studies on Reconstruction and Generation following standard protocol: 15 listeners rated randomly sampled audio clips on a 1-5 scale for overall quality (reported as mean±std).
>
> **Reconstruction MOS** (60 clips: 30 from AudioCaps, 30 from Song Describer Dataset):
>
> |Model|Latent Rate|MOS (↑)|
> |-|-|-|
> |Ground Truth|-|4.45±0.10|
> |**STAR-VAE (Ours)**|21.5 Hz|**4.32±0.12**|
> |$\epsilon$ar-VAE|43 Hz|4.21±0.14|
> |SAO|21.5 Hz|4.05±0.15|
>
> **Generation MOS** (60 clips from AudioCaps):
>
> |Model|MOS (↑)|
> |-|-|
> |Ground Truth|4.25±0.12|
> |**STAR-Gen (Ours)**|**3.92±0.16**|
> |TangoFlux|3.76±0.19|
> |SAO|3.71±0.18|
>
> At matched 21.5 Hz, STAR-VAE outperforms SAO in reconstruction MOS; STAR-Gen achieves the highest generation MOS among all baselines.  We will include MOS results and analysis in the revision of our manuscript.
>
> > Q1: Is it feasible to train $\epsilon$ar-VAE with STAR?
>
> Yes, STAR is **architecture-agnostic** and applicable to any continuous VAE exposing channel-wise KL regularization. However, $\epsilon$ar-VAE has not released training code, making reproduction infeasible. Additionally, it targets music-specific reconstruction with different optimization objectives (emphasizing phase accuracy), which may require re-tuning $\gamma$/$\beta$ ranges. We leave this as future work.
>
> > Q2: Why do STAR-Gen samples sound cleaner than direct STAR-VAE reconstructions?
>
> This observation is due to a fundamental difference between deterministic reconstruction and probabilistic generation:
>
> - **Reconstruction (STFT+GAN)**: The decoder directly inverts a *specific* compressed latent code to waveform via deterministic STFT inversion and GAN-based refinement. Compression artifacts (e.g., high-frequency loss at 21.5 Hz) are directly exposed in the output.
>
> - **Generation (Flow Matching)**: STAR-Gen learns to sample from the *latent data distribution* conditioned on text, not to invert individual codes. The flow model maps Gaussian noise to the latent manifold, producing samples that are *distributionally plausible* rather than point-wise reconstructions of specific inputs.
>
> In other words, the generator is not required to preserve compression artifacts from any particular input; instead, it learns the statistical structure of the latent space. The text condition and the LLM backbone further encourage semantically coherent outputs.

---

> > ### Author Rebuttal · Reviewer_Tf89 · 2026-04-02
> >
> > The authors addressed my comments, and the user study is particularly helpful.

---

> > > ### Author Response · Authors · 2026-04-02
> > >
> > > We sincerely thank the reviewer for updating the score to "Strong Accept" and confirming that all concerns have been fully resolved. We especially appreciate the recognition that the user study is particularly helpful, and we will incorporate the MOS evaluation results clearly into the revised manuscript.

---

### Official Review · Reviewer_rrCC · 2026-03-12

**Soundness:** 1
**Presentation:** 1
**Significance:** 2
**Originality:** 3
**Overall Recommendation:** 2
**Confidence:** 4

**Summary:**

The authors propose STAR, which weighs the KL divergence across the dimensions of a VAE bottleneck non-uniformly. This is applied to variational autoencoders with different architectures (CNN, Transformer, Mamba) on an audio reconstruction task, as well as a text-to-audio generative task.

**Compliance With Llm Reviewing Policy:**

Affirmed.

**Final Justification:**

The authors have not convinced me that the updated language they claim will end up in the final manuscript will clear up my misgivings regarding the natures of their claims and the accuracy of their paper. As such I will not be updating my rating.

**Key Questions For Authors:**

What body of literature can be cited to strengthen the issues pointed out in the Soundness concerns written above?

**Limitations:**

Yes

**Strengths And Weaknesses:**

Soundness:

I am concerned about the soundness of this paper, as much of the argumentation contains little relevant citations and evaluations.

The crux of this paper rests on the authors’ claims regarding the hierarchical structure of audio.
- The authors first base their claim on a diagram of a cat in Figure 1.
- There is no formal citation for the “problem of disordered information packing.”
- There is no citation for “reconstruction drift.”
- The statement “All latent channels possess equal capacity and semantic relevance” in Section 2.2 misunderstands that the fundamental promise of VAEs is to disentangle latent factors into orthogonal latent dimensions.
- There is not a single citation nor mathematical proof in section 2.3, where the authors “formalize” their claims.
- The authors conflate latent dimension in a VAE with frequency in audio signals throughout section 3.1. They also muddle the difference between reconstructing audio to maintain fine-grained acoustic detail in the signal vs. maintaining perceptual information.
- Figures 3a and 3b are relatively meaningless given that VAEs without STAR make no claims about information ordering.

By definition, reconstruction error should be assumed to increase when individual terms of the KL divergence are decreased via STAR. I think an improved evaluation would include a standard autoencoder as a baseline, or a VAE trained with cyclic annealing.

The linear probing in Section 4.3 does not cite what dataset is used nor describe the task in any relevant detail.


Presentation:

This paper is incredibly difficult to read due to odd choices in typesetting (overuse of bolding and italics), as well as overuse of jargon. Due to this, it is difficult to parse what the authors have proposed in their work. Figures present irrelevant and poorly supported information.

Significance:

The modest improvement in the provided metrics compared to the SAO baseline suggest this is an interesting line of inquiry, but requires more rigorous technical description and mathematical foundations.


Originality:

The paper proposes a regularization to VAE bottlenecks, which is underexplored in the literature to the best of my knowledge. Prior approaches have tried to estimate the latent dimension of a dataset, but not quite apply a weighting to individual dimensions of the bottleneck (Liu, Yue, et al. "Improving disentanglement in variational auto-encoders via feature imbalance-informed dimension weighting." Knowledge-Based Systems 296 (2024): 111818.)

---

> ### Author Rebuttal · Authors · 2026-03-30
>
> We sincerely appreciate the reviewer's recognition of STAR's originality (*"underexplored in the literature"*). We hope our point-by-point response can fully resolve all your concerns.
>
> >Figure 1 is based on a diagram of a cat
>
> As stated in the caption, Figure 1 uses a "metaphor" for conceptual illustration—standard practice for motivation figures. Reviewer vA6B confirms *"well-motivated and clearly illustrated (Figure 1)"*, and Reviewer 3v9B states Section 2 presents a *"clear and detailed explanation"*.
>
> >No citation for "disordered information packing"
>
> This is a **descriptive term we introduce** in Section 2.3 to characterize the observed consequence of Rate-Distortion-Regularity Trilemma, empirically evidenced by Figure 3(a). The problem formulation is detailed in Section 2.1-2.3, where we explain how isotropic priors lead to unstructured information allocation. We will cite related *but notably different* concepts: posterior collapse (Alemi et al., 2017; He et al., 2019) and perception–distortion trade-off (Blau & Michaeli, 2018).
>
> >No citation for "reconstruction drift"
>
> This is also a **descriptive term we introduce** in Lines 82-92 to characterize empirical findings evidenced in Table 1: adding Mamba to CNN under isotropic KL increases capacity but worsens spectral metrics. Please refer to our response to Reviewer vA6B on W4.
>
> >Section 2.2 misunderstands VAE disentanglement
>
> We respectfully clarify: our claim is not about disentanglement; we state that the isotropic prior applies **equal regularization pressure** across all channels, providing no inductive bias for capacity ordering. As in Lines 161-164, with the isotropic prior, the KL optimization landscape is "flat" w.r.t. the channel index $c$, penalizing information storage identically across all channels. We will revise Lines 160-161 to avoid confusion.
>
> >Section 2.3 "formalize" without proof
>
> We use "formalize" to mean **principled problem framing** based on information-theoretic reasoning, not theorem-level proof—standard for empirically-validated methodology papers. Reviewer Tf89 notes *"the theoretical reasoning is sound"* and Reviewer 3v9B gives Soundness: 4 (excellent). We will add citations to rate–distortion theory (Alemi et al., 2017) and natural sound statistics (Lewicki, 2002).
>
> >Conflate latent dimension with frequency; perceptual information
>
> We respectfully note two misunderstandings. (1) The **spectral hierarchy of audio** (low-frequency: structured, high-frequency: stochastic) serves as our **design motivation** (Lines 199-207), inspiring the capacity gradient concept. However, in the STAR formulation in Section 3.1, we describe latent channels in terms of **entropy** (High-Capacity Zone for Low-Entropy structure, Low-Capacity Zone for High-Entropy texture), **not** frequency. We did not claim latent channel $c$ maps to frequency band.
> (2) Per Lines 199-222, we clearly emphasize reconstructing both structural components and textural details for audio perceptual quality. STAR preserves semantically salient content for downstream generation, not only acoustic details, empirically reflected in better reconstruction, stronger probing, and consistent generation gains.
>
> >Figures 3a,b are meaningless
>
> We respectfully disagree. Precisely because standard VAEs with isotropic KL cannot enforce information ordering, Figure 3a,b provide diagnostic comparisons. Reviewer vA6B states *"the 'PCA-like energy compaction' shown in the truncation analysis (Figure 3b) provides **compelling evidence that STAR achieves its goal of information sorting**"*, and Reviewer Tf89 confirms *"the empirical evidence presented in the channel-wise information rate (3a) and truncation analyses (3b) **strongly supports the authors' claims**"*.
>
> >AE or cyclic annealing baseline
>
> The reviewer states *"reconstruction error should increase when KL is decreased"*; however, lower KL penalty allows more information to be encoded, typically improving reconstruction. Crucially, **our goal is not reconstruction alone but also downstream generation**: STAR creates a structured latent topology that directly benefits flow-matching priors (Table 2). Reconstruction-focused AE and VAE trained with cyclic annealing are not suitable baselines.
>
> >Linear probing lacks dataset/task description
>
> Task setup is in Appendix C.4. Evaluation is on AudioCaps test set.
>
> >Modest improvement over SAO baseline
>
> We respectfully disagree. Reviewer 3v9B noted *"significant improvements over strong baselines"* on reconstruction and *"clear gains"* on generation; Reviewer vA6B recognized *"significant improvements in text-to-audio metrics (FDopenl3, CLAP) compared to traditional diffusion models"*; Reviewer Tf89 observed that STAR-VAE *"substantially increases SAO's downstream generative performance across all metrics"*.
>
> >Presentation
>
> Reviewers vA6B, Tf89, and 3v9B all rate Presentation as 3 (good). While readability preferences vary, we will reduce formatting density where possible.

---

> > ### Author Rebuttal · Reviewer_rrCC · 2026-04-03
> >
> > Fundamentally, the writing of this paper is muddled and difficult for a human to read.
> >
> > Regarding the "metaphor" of the cat diagram, I maintain that this is a poor framing for the paper given that the evaluation presented in the paper spans speech, sound, and music generation. Throughout the paper, the "hierarchical structure of audio" may refer to the compressibility of low frequency sounds vs high frequency sounds, constituent components of speech that influence intelligibility, or perceptual qualities of an acoustic scene that split on foreground/background (fishbowl breaking vs splash of water afterwards). There is no clear definition nor citation anywhere in the paper to clarify this.
> >
> > The authors demonstrate that reconstruction performance improves in a structured manner as dimensions in the latent space are retained (i.e. not truncated) under their proposed weighting scheme. This is an opportunity for the authors to demonstrate what hierarchy of audio is demonstrated in this process: do low frequency sounds emerge first? formant traces in speech? foreground occurrences in audio scenes? Some level of human evaluation of the outputs of these models would go a long way to clarify what is happening in this framework.
> >
> > In a similar vein, I still think it reasonable to compare to a standard AE to demonstrate that there is value added when penalizing latent embeddings at all. Apologies for my human typo regarding KL divergence and reconstruction performance. In my experience, lowering the beta value in a beta-VAE improves reconstruction performance.
> >
> > I still maintain that the formalization of "the Rate-Distortion-Regularity Trilemma in continuous audio VAEs" is no formalization at all, and instead borrows jargon from information theory and stochastics with little theoretical or mathematical basis.
> >
> > It is clear that good engineering work has gone into this paper, as demonstrated by the objective metrics. However, in my personal estimation, the lack of scientific and mathematical rigor when formalizing an information-theoretic conjecture, lack of clear descriptive language across diagrams/figures/appendices, and lack of human evaluation make it unfit for publication.
> >
> > As such I will not be updating my score.

---

> > > ### Author Response · Authors · 2026-04-04
> > >
> > > We thank the reviewer for the follow-up. We provide the following clarifications to ensure that the paper and our original rebuttal are understood accurately.
> > >
> > > **1. On human evaluation.**
> > > We would like to clarify that **we have already provided a comprehensive human evaluation MOS study in our original rebuttal, specifically in our response to Reviewer Tf89’s W2, covering both reconstruction and downstream generation.** The MOS results demonstrate that at matched 21.5 Hz, STAR-VAE outperforms SAO in reconstruction MOS;  STAR-Gen achieves the highest generation MOS among all baselines.
> > > This study explicitly validates the perceptual advantages of STAR-VAE and STAR-Gen. Notably, **this exact study successfully resolved the concerns of Reviewer Tf89, who noted that "the user study is particularly helpful".** We will ensure that these MOS results and analysis are included in the revised manuscript.
> > >
> > > **2. On the meaning of “hierarchy of audio.”**
> > > Our claim is that **STAR induces ordered allocation in latent space by information entropy (Sec 3.1)**.  As we emphasized in our original rebuttal to the reviewer, we do **not** claim a one-to-one mapping between latent channel index and a specific physical frequency band or acoustic event category. Accordingly, the truncation analysis in Figure 3(b) is intended to demonstrate **ordered information compaction**, i.e., that lower-index channels under STAR contain more reconstruction-critical information, rather than to establish a direct latent-to-physical-feature correspondence.
> > >
> > > We also wish to clarify that the paper does not leave this notion undefined. **Section 3.1 defines the hierarchy in terms of information density / entropy structure and cites prior work on the heavy-tailed statistics of natural signals (Field, 1987; Voss & Clarke, 1975).** We agree that the wording can be made clearer in revision, but the paper does provide both a definition and citations for the intended motivation. **The suggested question of which specific physical cues emerge first under truncation is therefore an additional interpretability analysis, rather than a prerequisite for defining the claim made by STAR.**
> > >
> > > **3. On the AE baseline.**
> > > Our position remains that the central goal of this paper is **not unconstrained audio reconstruction in isolation**, but the design of a **continuous latent tokenizer for downstream generative models**, where latent regularity is part of the objective. **A deterministic AE removes the latent regularization mechanism under study and therefore does not evaluate the same problem setting as STAR.** For this reason, we compared against strong continuous VAE tokenizers such as SAO and $\epsilon$ar-VAE. An AE could still be used as an additional reconstruction-oriented reference, but we do not view it as a necessary baseline for assessing the main contribution of this work.
> > >
> > > **4. On the “formalization” of the trilemma.**
> > > We would like to clarify that **our use of “formalization” refers to problem formulation, not theorem-level proof.** This is also reflected in the actual section title, “Preliminaries and Problem Formulation.” The role of Section 2 is to structure the empirical tension among rate, reconstruction fidelity, and latent regularity in continuous audio VAEs. This framing is then supported by the analyses in the paper: ordered vs. disordered information allocation is examined through channel-wise KL patterns and truncation behavior, and reconstruction drift is evidenced by the degradation in spectral metrics when adding Mamba without STAR. If the word “formalization” is interpreted too strongly, we are happy to revise that wording in the paper; however, we would like to distinguish this wording issue from the empirical phenomena demonstrated by the experiments.
> > >
> > > **5. On writing, framing, and the role of the proposed contribution.** \
> > > We understand that the current presentation may not work equally well for all readers. In the initial review, the presentation concern was described in part as arising from **“odd choices in typesetting (overuse of bolding and italics)”**, and we already committed in the rebuttal to reducing this formatting density in the revision.
> > >
> > > As explained in our original rebuttal to the reviewer, our intention with Figure 1 was as a **conceptual motivation** using a metaphor, not as evidentiary support for the technical claims. At the same time, we would like to emphasize that the contribution of the paper is not the metaphor itself, but the proposed **channel-wise topology-aware regularization**, the effect of which is reflected in the quantitative analyses, ablations, and downstream generation results. This is also consistent with the reviewer’s initial recognition that weighting individual dimensions of the bottleneck is **“underexplored in the literature.”** We will refine the framing and terminology further in revision so that the novelty and empirical contribution are conveyed more clearly.

---

### Official Review · Reviewer_vA6B · 2026-03-13

**Soundness:** 2
**Presentation:** 3
**Significance:** 2
**Originality:** 3
**Overall Recommendation:** 4
**Confidence:** 3

**Summary:**

This paper addresses the "Rate-Distortion-Regularity (R-D-R) Trilemma" in continuous Variational Autoencoders (VAEs) for audio modeling. The authors identify a topological mismatch between the isotropic Gaussian prior used in standard VAEs and the intrinsic spectral hierarchy of audio signals. This paper strives to assess a central concept of latent space geometry by introducing Structured Topology-Aware Regularization (STAR), which uses a Gamma-Growth function to impose a capacity gradient across latent channels. This approach routes structured global information to high-capacity channels and stochastic textures to low-capacity ones. The authors further propose **STAR-VAE**, a hybrid CNN-Mamba architecture, and **STAR-Gen**, an LLM-based Flow Matching framework, to validate the effectiveness of the structured latent space.

**Compliance With Llm Reviewing Policy:**

Affirmed.

**Key Questions For Authors:**

Have you attempted to train STAR-VAE directly from Phase I using the Gamma-Growth penalty? Does the "Safe Harbor" require an initial isotropic phase to stabilize?

**Strengths And Weaknesses:**

## **Strengths**
*   **Intuitive Motivation:** The concept of a "Capacity Gradient" to match the natural power-law decay of audio signals is well-motivated and clearly illustrated (Figure 1).
*   **Methodological Versatility:** STAR is shown to be effective across different architectures (CNN-only and hybrid), suggesting broad applicability as a general tokenizer.
*   **Empirical Results:** This study aims to examine a central concept of latent ordering; the "PCA-like energy compaction" shown in the truncation analysis (Figure 3b) provides compelling evidence that STAR achieves its goal of information sorting.
*   **Generative Quality:** The integration with Flow Matching (STAR-Gen) shows significant improvements in text-to-audio metrics (FDopenl3, CLAP) compared to traditional diffusion models.

## **Weaknesses**
*   **Hyperparameter Sensitivity:** The performance seems heavily dependent on the choice of the Gamma-Growth parameter $\gamma$. While $\gamma=2.0$ is optimal for the tested datasets, there is limited discussion on how this generalizes to vastly different audio domains (e.g., highly impulsive vs. purely tonal sounds).
*   **Training Complexity:** The two-stage training strategy (Phase I isotropic pre-training followed by Phase II structured fine-tuning) adds complexity to the pipeline. It is unclear if the model can be trained from scratch using STAR effectively.
*   **Baseline Scope:** While the paper compares against several continuous VAEs, a more rigorous comparison against high-bitrate discrete neural codecs (e.g., DAC) in terms of "bits-per-second vs. quality" would help clarify the efficiency gains of this continuous approach.
*   **Ablation Detail:** The "Reconstruction Drift" phenomenon is a key claim for the necessity of STAR in high-capacity models like Mamba, but the quantitative delta in spectral metrics between "Hybrid w/o STAR" and "CNN-VAE" is relatively small, making the "drift" argument feel slightly overstated.

---

> ### Author Rebuttal · Authors · 2026-03-30
>
> We sincerely thank the reviewer for the positive assessment and for recognizing STAR's *"intuitive motivation," "methodological versatility," "compelling evidence"* for information sorting, and *"generative quality".* We hope the following point-by-point response could resolve all your questions.
>
> > W1: Hyperparameter sensitivity of $\gamma$ across audio domains
>
> In STAR, $\gamma$ is a **shape hyperparameter** controlling capacity distribution. It determines (i) the proportion of low-penalty channels for global structure, and (ii) how sharply the bottleneck transitions to high-penalty regions. A too small $\gamma$ leads to structured information competing with high-entropy detail prematurely; a too large $\gamma$ weakens channel ordering. A **moderately convex** schedule ($\gamma \approx 2$) balances these effects (Section 3.1, Appendix C.1).
>
> **Cross-audio-domain Generalizability:** Besides the optimal $\gamma=2.0$ observed on AudioCaps, experiments on the vastly diverse **VGGSound test set** (309 event categories, ~15000 samples) verify that **the optimal** $\gamma=2.0$ **generalizes to acoustically diverse domains**, confirming that convex allocation shape—**not dataset-specific tuning**—is key to STAR's effectiveness.
>
> |Dataset|Metric|$\gamma$=1.5|$\gamma$=2.0|$\gamma$=3.0|
> |-|-|-|-|-|
> |AudioCaps|FAD↓|2.45|**2.31**|2.52|
> |AudioCaps|STFT-D↓|1.32|**1.17**|1.25|
> |VGGSound|FAD↓|1.72|**1.57**|1.65|
> |VGGSound|STFT-D↓|1.92|**1.84**|1.88|
>
> > W2 and Question: Two-stage training complexity / Can STAR be trained from scratch?
>
> Audio VAE training (Section 2.1) suffers from a **common multi-objective entanglement problem**: the simultaneous optimization of spectral reconstruction (STFT), adversarial fidelity (GAN), and latent regularization (KL) creates conflicting gradients (Yu et al., "Gradient Surgery for Multi-Task Learning", NeurIPS 2020). Hence, when STAR's channel-dependent KL weighting is applied from scratch, the model struggles to balance reconstruction with structured latent utilization before a stable manifold is established. In contrast, our two-stage strategy addresses this: Phase I establishes a stable reconstruction manifold under isotropic KL; Phase II reorganizes information density via STAR without disrupting learned reconstruction.
>
> **Comparison on AudioCaps:**
>
> |Setting|STFT-D↓|MSD↓|FAD↓|SI-SDR↑|
> |-|-|-|-|-|
> |STAR from scratch|1.28|0.88|2.78|-0.67|
> |STAR two-stage|**1.17**|**0.75**|**2.31**|**-0.03**|
>
> **Findings:** Two-stage training outperforms training STAR from scratch. The overhead is modest: Phase I (\~150h) + Phase II (\~70h), with Phase II accounting for only ~32% of total time. For reference, SAO also adopts two-stage training (183h encoder + 273h decoder-only on 32 A100s).
>
> > W3: Comparison with discrete codecs (bits-per-second vs. quality)
>
> We compare STAR-VAE with high-bitrate discrete codecs on AudioCaps:
>
> |Model|Type|Sample Rate|Latent Rate|STFT-D↓|MSD↓|FAD↓|
> |-|-|-|-|-|-|-|
> |EnCodec|Discrete|32k (mono)|50 Hz|1.15|0.70|**0.98**|
> |DAC|Discrete|44.1k (mono)|86 Hz|1.10|0.70|1.55|
> |STAR-VAE|Continuous|44.1k (stereo)|43 Hz|**1.01**|**0.63**|1.56|
> |STAR-VAE|Continuous|44.1k (stereo)|21.5 Hz|1.17|0.75|2.31|
>
> **Findings:** STAR-VAE (43 Hz) achieves the best STFT-D (1.01) and MSD (0.63) at half the latent rate of DAC (86 Hz), while handling stereo audio. This shows the efficiency gain of our continuous approach—structured latent organization yields better spectral fidelity per unit of latent capacity. At 21.5 Hz, reconstruction remains competitive while enabling direct flow-matching generation without AR stages. We will include the above experimental comparisons in the revision of our manuscript.
>
> > W4: Quantitative Delta for "Reconstruction Drift"
>
> As defined in Lines 82-92, **Reconstruction Drift** describes a counter-intuitive phenomenon: adding Mamba layers to the CNN encoder (Hybrid w/o STAR) increases model capacity and enables global context modeling, which *should* improve reconstruction—yet Table 1 shows the opposite. Hybrid w/o STAR yields *better* FAD (2.74 vs. 3.36) but *worse* spectral metrics (STFT-D: 1.35 vs. 1.28, SI-SDR: -1.43 vs. -1.14) compared to CNN-VAE.
>
> This split—*improved semantics but degraded spectral fidelity*—is precisely what we define as Reconstruction Drift: additional capacity under isotropic KL does not translate to better fine-grained reconstruction.
>
> Crucially, after applying STAR, STAR-VAE achieves the best results **across *all* metrics** (STFT-D: 1.17, FAD: 2.31, SI-SDR: -0.03), demonstrating that STAR successfully unlocks Mamba's potential. This improvement further confirms the drift phenomenon—without structured regularization, the added capacity is misallocated.

---

> > ### Author Rebuttal · Reviewer_vA6B · 2026-04-04
> >
> > After reviewing the authors' rebuttal and carefully considering the feedback from other reviewers, I have decided to maintain my original score. While the rebuttal was somewhat limited in addressing all concerns, the collective discussion and the current state of the paper justify my initial assessment.

---

> > > ### Author Response · Authors · 2026-04-04
> > >
> > > Thank you very much for your continued engagement with our paper and for maintaining your positive assessment of the work.
> > >
> > > In our rebuttal, we provided additional empirical evidence and clarifications to **address all the concerns raised in your original review**. In particular, we provided:
> > >
> > > (1) **cross-domain validation of the $\gamma$ parameter on the vastly acoustically diverse VGGSound benchmark**, showing that the preferred convex allocation and the optimal $\gamma$ generalize beyond AudioCaps;
> > > (2) a direct **STAR from-scratch vs. two-stage training comparison**, confirming that the isotropic warm-up is beneficial for stable optimization;
> > > (3) new **comparisons against high-bitrate discrete codecs** (including DAC and EnCodec), clarifying the bitrate-quality tradeoff of the continuous STAR-VAE approach and showing the efficiency gain of our continuous approach—structured latent organization yields better spectral fidelity per unit of latent capacity; \
> > > (4) we also further clarified the intended meaning of the "reconstruction drift" phenomenon in the hybrid backbone setting.
> > >
> > > If there are **specific remaining concerns** that you believe were not fully addressed in our rebuttal, we would be very happy to clarify them further. In any case, we will incorporate all of the above **new experiments, analyses, and clarifications** into the **revised manuscript** to make the paper clearer and more complete.

---

### Decision · Program_Chairs · 2026-04-30

**Decision:**

Accept (regular)

**Comment:**

While one reviewer raised concerns regarding the scope and clarity of the contributions, the other three reviewers found the paper to be technically sound and positively evaluated its contributions. The authors’ rebuttal adequately addressed the major points. Considering the overall balance of reviews, I recommend acceptance.